# Audio-Sync Video Generation
# with Multi-Stream Temporal Control

**Shuchen Weng**[1†]   **Haojie Zheng**[1,2†]   **Zheng Chang**[3]   **Si Li**[3]   **Boxin Shi**[4,5‡]   **Xinlong Wang**[1‡]

[1]Beijing Academy of Artificial Intelligence
[2]School of Software and Microelectronics, Peking University
[3]School of Artificial Intelligence, Beijing University of Posts and Telecommunications
[4]State Key Lab of Multimedia Info. Processing, School of Computer Science, Peking University
[5]Nat'l Eng. Research Ctr. of Visual Tech., School of Computer Science, Peking University
{scweng, wangxinlong}@baai.ac.cn,  suimu@stu.pku.edu.cn
{zhengchang98,lisi}@bupt.edu.cn,  shiboxin@pku.edu.cn

## Abstract

Audio is inherently temporal and closely synchronized with the visual world, making it a naturally aligned and expressive control signal for controllable video generation (*e.g.*, movies). Beyond control, directly translating audio into video is essential for understanding and visualizing rich audio narratives (*e.g.*, Podcasts or historical recordings). However, existing approaches fall short in generating high-quality videos with precise audio-visual synchronization, especially across diverse and complex audio types. In this work, we introduce MTV, a versatile framework for audio-sync video generation. MTV explicitly separates audios into speech, effects, and music tracks, enabling disentangled control over lip motion, event timing, and visual mood, respectively—resulting in fine-grained and semantically aligned video generation. To support the framework, we additionally present DEMIX, a dataset comprising high-quality cinematic videos and demixed audio tracks. DEMIX is structured into five overlapped subsets, enabling scalable multi-stage training for diverse generation scenarios. Extensive experiments demonstrate that MTV achieves state-of-the-art performance across six standard metrics spanning video quality, text-video consistency, and audio-video alignment. Project page: https://hjzheng.net/projects/MTV/.

## 1 Introduction

Audio is a fundamental medium in daily life, crucial for both information delivery (*e.g.*, communication, notifications, and education) and immersive experiences (*e.g.*, enhancing the impact of film visuals). Despite the prevalence of audio-centric platforms (*e.g.*, Podcasts), content presented solely through audio lacks the visual dimension needed to fully convey the richness of events. Since audio is naturally temporal and inherently synchronized with the visual world, researchers [1–3] have devoted considerable attention to translating audios into corresponding videos to enhance audience understanding of rich audio narratives (*e.g.*, historical recordings).

Despite great progress, existing methods face practical limitations in generating high-fidelity cinematic videos with precise synchronization (*e.g.*, pouring water into the transparent cup), primarily due to: *(i)* **Under-specified audio-visual mapping.** Current approaches handle a wide spectrum of audios and map them to various target scenes (*e.g.*, landscapes [4], dancing [5], music performances [6]). This broad representation scope potentially leads to ambiguous mappings lacking specificity between audio and visual features. *(ii)* **Inaccurate temporal alignment.** Existing methods primarily focus on

---

† Equal contributions. ‡ Corresponding authors.

39th Conference on Neural Information Processing Systems (NeurIPS 2025).

**Character-centric narrative** *... A man in a brown jacket and a blue shirt ... talking on a mobile phone ...*

**Multi-character interaction** *... A male on the left, wearing a black suit ... A woman on the right, wearing a red dress ...*

**Sound-triggered events** *... The focus is on the water's movement as it is poured into the glass from above ...*

**Music-shaped ambiance** *... A young woman ... flowers with purple and yellow ... A man dressed in a dark jacket ...*

**Camera movement** *... An old, rusted car driving on a suburban street ... with a faded white paint job ...*

Figure 1: MTV demonstrates versatile audio-sync video generation capabilities following user-provided text descriptions specifying scenes and subjects. Capabilities shown include producing videos centered on targeted characters (1st and 2nd rows) while triggering events with sound effects (3rd row), generating visual mood with accompanying music (4th row), and adaptively handling camera movement (5th row). We present these generated videos in the supplementary materials.

building scene-level semantic consistency (*e.g.*, translating engine sound to a car-centered video), struggling with accurate timing correspondence between individual audio events and their visual features (*e.g.*, speech [7], motion [8], and visual mood [9]).

In this paper, we propose the **MTV** framework, enabling **M**ulti-stream **T**emporal control for audio-sync **V**ideo generation to overcome aforementioned issues, with versatile capabilities across scenarios illustrated in Fig. 1. Instead of attempting a direct mapping from composite audios, we explicitly separate audios into distinct controlling tracks (*i.e.*, speech, effects, and music), inspired by CDX'23[1]. To provide sufficient high-quality video clips with demixed audio tracks, we contribute a large-scale DEMIX dataset with tailored data processing, including 392K video clips with 1.2K hours. These tracks enable the model to precisely control lip motion, event timing, and visual mood, resolving the ambiguous mapping. To further incorporate rich visual semantics beyond direct audio cues, we leverage features (*e.g.*, subject gesture, scene appearance, camera movement) initially derived from a pretrained text-to-video model [10], and subsequently finetuned using video clips from the DEMIX dataset. To enable the progressive extension of learned high-level video semantic features stage-by-stage, this dataset is structured into five overlapped subsets. A multi-stage training strategy is introduced to learn concrete and localized controls (*e.g.*, lip motion) towards more abstract and global influences (*e.g.*, visual mood), leading to clear audio-visual relationships.

---

[1]https://www.aicrowd.com/challenges/sound-demixing-challenge-2023

To achieve accurate temporal alignment, we propose the Multi-Stream Temporal ControlNet (MST-ControlNet) within the MTV framework. The interval stream is designed for specific feature synchronization, which extracts features from the speech and effects tracks. It employs interval interaction blocks to understand each track individually and construct their interplay, maintaining the coherence with inferred semantic features. After that, interval feature injection module inserts features of each track into corresponding time intervals to drive lip motion and event timing. Since visual mood typically covers the entire video clip, the holistic stream is designed for overall aesthetic presentation, which extracts features from the music track using the holistic context encoder. These features then serve as style embeddings, applied uniformly to all frames through global style injection, controlling the visual mood.

We summarize our contributions as follows:

- We present MTV, a versatile audio-sync video generation framework by demixing audio inputs, achieving precise audio-visual mapping and accurate temporal alignment.
- We introduce an audio-sync video generation dataset structured into five overlapped subsets, presenting the multi-stage training strategy for learning audio-visual relationships.
- We propose the multi-stream temporal ControlNet to distinctively process demixed audio tracks and precisely control lip motion, event timing, and visual mood, respectively.

## 2 Related Works

### 2.1 Video Diffusion Model

The field of video generation has made significant progress with the adoption of diffusion models. Early approaches [11–13] extend the dynamic modeling capabilities of pretrained text-to-image diffusion models [14] by incorporating temporal layers (*e.g.*, 3D convolutions [15] and temporal attention [16]). However, these methods face inherent challenges in capturing long-range spatial-temporal dependencies due to the convolutional architectures of their backbone (*e.g.*, UNet [17]). To overcome this limitation, Sora report [18] presents the potential of the diffusion transformer (DiT) [19] architecture, prompting a shift towards integrating 3D VAE [20] for spatial-temporal compression and scaling up to train the entire DiT-based model. Further improvement has been achieved by recent foundation models through adaptive layernorm modules [10], progressive scaling [21, 22], and post-training techniques [23]. These advancements in text-to-video models provide a strong foundation and powerful generative priors that could potentially be leveraged for related cross-modal tasks, such as high-quality audio-sync video generation.

### 2.2 Audio-driven Image Animation

Audio-driven image animation aims to generate dynamic visuals from a static image, synchronized with user-provided audios. Several previous works animate general objects or scenes while maintaining audio-visual consistency. Sound2Sight [24] and CCVS [25] leverage the context of preceding frames to achieve audio-driven subsequent frames generation. TPOS [26] uses audios with variable temporal semantics and amplitude to guide the denoising process. ASVA [27] incorporates a temporal audio control module for effective audio synchronization. Other works concentrate on audio-driven human animation. Talking head [7, 28–30] focus on animating human face images to produce lip motion that synchronize with the speech. Recent works extend animation beyond the head to include half-body movements [31] and introduce pose control for full-body animation [32]. Another specific application is music-to-dance [33, 34], which generates human dance according to the beat of the music. Despite the audio-visual synchronization of these methods, their reliance on static images restricts models' capability to generate dynamic scenes required for cinematic videos.

### 2.3 Audio-sync Video Generation

Audio-sync video generation does not require additional images for reference, offering the potential for free scene creation. Early works are designed based on VQGAN [35] and StyleGAN [36], achieving audio control through multi-modal autoregressive transformers [2] and style code alignment [4, 37]. Recently, following the success of diffusion models demonstrating effectiveness in general video generation, researchers have turned their attention. Highlighting the benefit of multi-modal

Table 1: Comparison of DEMIX dataset and previous datasets.

| Method | Year | Modality | | Scene | | | Audio component | | | | Specifications | |
|---|---|---|---|---|---|---|---|---|---|---|---|---|
| | | Text | Audio | People | Objects | Cinematic | Speech | Effects | Music | Demix | Clips | Hours |
| UCF-101 [38] | 2012 | – | ✓ | ✓ | – | – | – | ✓ | ✓ | – | 13K | 27 |
| HIMV-200K [39] | 2017 | – | ✓ | ✓ | ✓ | ✓ | – | – | ✓ | – | 200K | – |
| AudioSet [40] | 2017 | – | ✓ | ✓ | ✓ | – | ✓ | ✓ | ✓ | – | 2.1M | 5.8K |
| VoxCeleb2 [41] | 2018 | – | ✓ | ✓ | – | – | ✓ | – | – | – | 150K | 2.4K |
| VGGSound [42] | 2020 | – | ✓ | ✓ | ✓ | – | ✓ | ✓ | ✓ | – | 200K | 550 |
| WebVid-10M [43] | 2021 | ✓ | – | ✓ | ✓ | – | – | – | – | – | 10.7M | 52K |
| Landscape [4] | 2022 | – | ✓ | – | ✓ | – | – | ✓ | – | – | 9K | 26 |
| InternVid [44] | 2024 | ✓ | ✓ | ✓ | ✓ | – | ✓ | ✓ | ✓ | – | 7.1M | 760K |
| Ours (DEMIX) | 2025 | ✓ | ✓ | ✓ | ✓ | ✓ | ✓ | ✓ | ✓ | ✓ | 392K | 1.2K |

conditions, TA2V [6] demonstrates that conditioning on both text descriptions and audio inputs significantly enhances the quality of generated videos. To achieve audio-visual alignment at both global and temporal levels, TempoTokens [1] designs a lightweight adapter for text-to-video generation model. Introducing a unified diffusion architecture, MM-Diffusion [5] enables both joint audio-video generation and zero-shot audio-sync video generation. Leveraging diffusion-based latent aligners for open-domain audio-visual generation, Xing *et al.*[3] achieve the audio-sync video editing and open-domain content creation. Although great progress has been made, audio-sync video generation still faces under-specific audio-visual mapping and inaccurate temporal alignment. Therefore, achieving cinematic quality remains challenging.

## 3 Dataset

We introduce the DEMIX dataset, tailored for training demixed audio-sync video generation models.

**Data source.** The training data is sourced from three aspects: *(i)* 65 hours of talking head videos from CelebV-HQ [45]; *(ii)* 4,923 hours of cinematic videos from MovieBench [46] (69h), Condensed Movies [47] (1,270h), and Short-Films 20K [48] (3,584h); and *(iii)* 8,903 hours film-related videos from YouTube. All collected videos include their accompanying audio tracks.

**Video filtering.** Following previous video generation models [10, 12, 49], we use PySceneDetect [50] to segment video into single-shot clips. Audiobox-aesthetics [51] is further used to assess the quality of accompanying audio, removing clips with low scores. For the left video clips, we annotate each one with text descriptions using LLaVA-Video [52].

**Demixing filtering.** To improve audio demixing reliability, we employ a dual-demixing comparison strategy, comparing demixing outputs from MVSEP [53] (speech, effects, music) and Spleeter [54] (speech, others). After that, we calculate the L1 distance between the speech tracks. Next, the 'others' track from Spleeter is conditionally compared: to the effects track from MVSEP if music is silent (below -45dB), and to the music track if effects are silent. Clips are discarded only if high L1 distances are found on any of the comparable pairs.

**Voice-over filtering.** To build clear audio-visual relationships for cinematic videos, we first detect whether people are present in the videos using YOLO [55]. Next, we perform speaker diarization for the accompanying audio using Scribe [56] to identify active speaker segments and count the number of speakers. After that, we detect the active speaker from videos for each frame using TalkNet [57]. As a result, we can discard clips where speech occurs in the audio but the video analysis detects neither a visible person nor an active speaker in the corresponding frames.

**Subset division.** To facilitate multi-stage training for versatile audio-sync video generation models, the filtered DEMIX data is structured into five overlapped subsets. The basic face subset comprises all talking head videos. The remaining cinematic and film-related videos are then categorized to form the other subsets: assignment to single character or multiple characters depends on the annotated human count, while assignment to sound event or visual mood occurs if the respective effects or music track is non-silent.

**Data statistics.** After data collection and filtering, our DEMIX dataset includes 18K basic face, 54K single character, 39K multiple characters, 166K sound event, and 195K visual mood data, tailored for

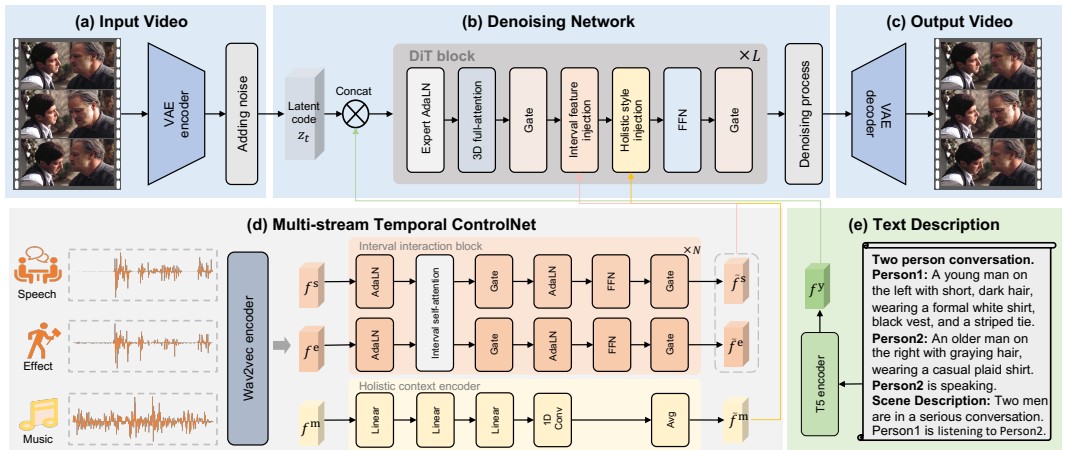

Figure 2: The pipeline of our MTV framework. (a-c) MTV is built on a pretrained text-to-video model [10] that provides strong generative priors for synthesizing diverse visual scenarios. (d) Explicitly separated audio tracks (*i.e.*, speech, effects, music) are fed into our proposed multi-stream temporal ControlNet to ensure synchronization for lip motion, event timing, and visual mood. (e) The MTV framework is trained on our contributed DEMIX dataset with five overlapped subsets and tailored text structures, enabling a multi-stage training strategy for audio-sync video generation.

cinematic videos, totaling non-overlapped 392K clips with 1.2K hours, accompanied by demixed audio tracks[2]. For comprehensive evaluation, we hold out 1K video clips from the dataset to form the testing set. We provide an additional comparison with existing audio-related datasets [4, 38–44] in Tab. 1, highlighting that ours is tailored for versatile audio-sync video generation using demixed audio tracks, while robustly covering scenarios with people, objects, and cinematic visuals.

## 4 Method

This section begins with an overview of our MTV framework for audio-sync video generation (Sec. 4.1). Next, we detail the Multi-stream Temporal ControlNet (MST-ControlNet), including the interval stream for specific feature synchronization, and the holistic stream for overall aesthetic presentation (Sec. 4.2). Finally, we present the multi-stage training strategy for effectively learning audio-visual relationships (Sec. 4.3).

### 4.1 Overview

MTV generates audio-sync videos based on user-provided text descriptions $y$ (specifying the scenes and subjects) and demixed audio tracks $a = \{a^{s}, a^{e}, a^{m}\}$ (representing speech, effects, and music) to respectively drive the lip motion, event timing, and visual mood. The pipeline is illustrated in Fig. 2.

**Video compression.** As presented in Fig. 2 (a), MTV is equipped with a pretrained spatio-temporal variational autoencoder (VAE) encoder $\mathcal{E}$ to map video clips $x$ into latent code $z_0 = \mathcal{E}(x)$. After that, its corresponding VAE decoder $\mathcal{D}$ is used to reconstruct video clips from the latent code $x = \mathcal{D}(z_0)$.

**Denoising network.** As presented in Fig. 2 (b), we concatenate the text embeddings $f^{y}$ and noised latent code $z_t$ before feeding them into the network to ensure the video-text correspondence. The expert Adaptive LayerNorm (AdaLN) [10] then independently processes text and video features within this unified sequence. Next, 3D full-attention is used to interact semantics of text embeddings with corresponding video features. After being extracted by MST-ControlNet, audio cues are integrated via the interval feature injection and holistic style injection mechanisms. Finally, a feed-forward network (FFN) is used to refine the resulting video features.

**Denoising process.** As presented in Fig. 2 (c), MTV finally generates audio-sync videos by iteratively denoising latent codes. During training, at each time step $t \in \{0, \ldots, T\}$, Gaussian noise $\epsilon_t \sim$

---

[2]Dataset samples are visualized in the supplementary materials.

$\mathcal{N}(0, 1)$ is added to the clean latent code $z_0$ to produce a noised latent code $z_t = \sqrt{\bar{\alpha}_t} z_0 + \sqrt{1 - \bar{\alpha}_t} \epsilon_t$. A diffusion transformer $\epsilon_\theta$ is trained to predict the noise $\epsilon_t$, given the noised latent code $z_t$, demixed audio tracks $a$, denoising time step $t$, and text descriptions $y$. The diffusion transformer is trained by minimizing the loss:

$$\mathcal{L}_{\text{dm}} = \mathbb{E}_{t, z_0, \epsilon_t \sim \mathcal{N}(0,1)} \left[ \| \epsilon_t - \epsilon_\theta(z_t, a, t, y) \|^2 \right]. \tag{1}$$

For inference, we iteratively denoise a randomly sampled noise $z_T \sim \mathcal{N}(0, 1)$ to obtain the latent code $z_0'$ to generate video clips with the VAE decoder $x' = \mathcal{D}(z_0')$.

### 4.2 Multi-stream Temporal ControlNet

After explicitly separating audios into speech, effects, and music tracks, we propose the MST-ControlNet to achieve accurate temporal alignment by respectively controlling lip motion, event timing, and visual mood. As presented in Fig. 2 (d), the architecture consists of an audio encoding module followed by two specialized streams.

**Audio encoding.** Given demixed audio tracks $a = \{a^s, a^e, a^m\}$, we initially extract their corresponding features $\{f^s, f^e, f^m\}$ from the demixed tracks using wav2vec [58]. After that, speech and effect features are fed into the interval stream for specific feature synchronization. Instead, music features are fed into the holistic stream for overall aesthetic presentation.

**Interval stream.** We design the interval stream to interval-wise control the lip motion and event timing. Specifically, we separately process speech features $f^s$ and effect features $f^e$ with a stack of linear layers and concatenate them before feeding them into $N$ interval interaction blocks. Within each block, these features are processed independently (via AdaLN, Gate, and FFN) to refine per-track understanding. To model their interplay at each time interval $i$, the corresponding speech features $f_i^s$ and effects features $f_i^e$ are jointly processed by a self-attention $[\tilde{f}_i^s, \tilde{f}_i^e] = \text{SelfAttn}([f_i^s, f_i^e])$. This interaction also maintains the coherence with inferred semantic features. Finally, interacted speech features $\tilde{f}^s$ and effects features $\tilde{f}^e$ are integrated into their corresponding time intervals via the interval feature injection mechanism:

$$h_i^s = \text{CrossAttn}(h_i, \tilde{f}_i^s), \quad h_i^e = \text{CrossAttn}(h_i, \tilde{f}_i^e), \tag{2}$$

where $h_i$ represents the video latent code at $i$-th interval. $\text{CrossAttn}(\cdot, \cdot)$ means a cross-attention, where the latent code serves as the query and the audio features as the key and value. Let $M$ be the number of intervals, the resulting latent code is then updated as $h' = \{h_i^s + h_i^e\}_{i=1}^M$.

**Holistic stream.** The holistic stream is designed to control the visual mood for the entire video clip. Specifically, we process the music features $f^m$ through a holistic context encoder, comprising three linear layers and a 1D convolutional layer to extract features representing the visual mood. Since the environmental ambiance typically covers the entire video clip, an average pooling is applied to merge all the intervals and transform them into holistic music features $\tilde{f}^m$. Next, these features are regarded as style embeddings. By independently transforming these features into scale factor $\gamma^m = \text{Linear}(\tilde{f}^m)$ and shift factor $\beta^m = \text{Linear}(\tilde{f}^m)$, we modulate the video latent code $h'$ uniformly across all intervals via the holistic style injection:

$$h^m = h' \odot (\gamma^m + 1) + \beta^m, \tag{3}$$

where $h^m$ is the modulated latent code, fed into the denoising network to refine video features.

### 4.3 Multi-stage training strategy

As the dataset is structured as five overlapped subsets, we introduce the multi-stage training strategy to progressively scale up the model stage-by-stage.

**Text structure.** As presented in Fig. 2 (e), we create a template to structure text descriptions, enabling our MTV framework to be compatible with these distinct training subsets. Specifically, this template begins with a sentence indicating the number of participants (*e.g.*, "Two person conversation"), based on Scribe [56] speaker counts. It then consists of subsequent entries for each individual, starting with a unique identifier (*e.g.*, *Person1*, *Person2*) followed by their respective appearance description. Following these individual entries, an explicit identifier for the currently active speaker is specified. Finally, a sentence provides an overall description of the scene. Notably, when there is no active speaker in the video, only the overall description will be provided.

Table 2: Quantitative experiment results of comparison and ablation. ↑ (↓) means higher (lower) is better. Throughout the paper, best performances are highlighted in **bold**.

| Method | FVD ↓ | Temp-C (%) ↑ | Text-C (%) ↑ | Audio-C (%) ↑ | Sync-C ↑ | Sync-D ↓ |
|---|---|---|---|---|---|---|
| | Comparison with state-of-the-art methods | | | | | |
| MM-Diffusion [5] | 879.77 | 94.15 | 15.61 | 5.43 | 1.53 | 11.21 |
| TempoTokens [1] | 795.88 | 93.13 | 24.68 | 6.71 | 1.45 | 10.48 |
| Xing *et al.* [3] | 805.23 | 93.30 | 24.51 | 7.30 | 1.55 | 10.50 |
| Ours (MTV) | **626.06** | **95.40** | **26.55** | **26.22** | **3.17** | **9.43** |
| | Ablation study | | | | | |
| *W/o* SE | 667.81 | 95.30 | 26.49 | 24.68 | 2.46 | 9.55 |
| *W/o* SI | 626.46 | 94.84 | 25.50 | 19.64 | 2.53 | 9.76 |
| *W/o* TB | 698.36 | 95.14 | 26.37 | 24.50 | 2.31 | 9.78 |

**Training schedule.** We train the model from concrete and localized controls towards more abstract and global influences. Initially, we train the model to learn lip motion using the basic face subset. It then learns human pose, scene appearance, and camera movement on the single character subset. To handle scenarios with multiple speakers, we subsequently train the model on the multiple characters subset. Following this, our training focus shifts to event timing and extending subject understanding from humans to objects using the sound event subset. Finally, we train the model on the environmental ambiance subset to improve its representation of visual mood.

**Training details.** We initialize our spatial-temporal VAE and DiT backbone with pretrained weights from CogVideoX [10] and train our model to generate audio-sync videos at a $480 \times 720$ resolution. For each stage, we train our model for 40K steps on 24 NVIDIA A800 GPUs using the Adam-based optimizer [59] with a learning rate of $1 \times 10^{-5}$, where MST-ControlNet and attention layers of the backbone are trainable. For inference, our model requires 280s to generate a 49-frame audio-sync video on a NVIDIA A100 GPU.

## 5 Experiments

### 5.1 Comparison with state-of-the-art methods

As audio-sync video generation is an emerging task, the relevant comparison methods are still developing. We compare our method with three recent state-of-the-art approaches in our DEMIX dataset. For TempoTokens [1] and Xing *et al.* [3], we evaluate them using both text descriptions and corresponding audios as their original configuration. Since MM-Diffusion [5] can only support audio inputs and its training focuses on specific landscape and dancing, we finetune it to ensure a fair comparison. 50 videos are randomly selected from the testing set for evaluation.

**Quantitative comparisons.** As presented in Tab. 2, we quantitatively evaluate performance across three main aspects: *(i)* Visual quality is assessed using Frechét Video Distance (FVD) [60]. *(ii)* Temporal consistency (Temp-C) is measured by calculating similarity between consecutive frames using CLIP [61]. *(iii)* We examine text-video alignment via Text Consistency (Text-C) [62], audio-video alignment using Audio Consistency (Audio-C) [63], and specifically lip motion synchronization with Sync-C and Sync-D [64]. As a result, our framework outperforms state-of-the-art methods across all six quantitative metrics. These metric details are provided in the supplementary materials.

**Qualitative comparisons.** As presented in Fig. 3, qualitative comparisons with state-of-the-art methods [1, 3, 5] highlight the advantages of our framework. For instance, even after finetuning MM-Diffusion [5] for over 320K steps using the official code on 8 NVIDIA A100 GPUs, it still struggles with generating cinematic videos. TempoTokens [1] struggles to generate cinematic videos for complex text-specified scenarios, resulting in unrealistic human expressions (Fig. 3 left). Xing *et al.* [3] find it difficult to effectively achieve audio synchronization for specific event timing, leading to incorrect rendering of human gestures for guitar performance (Fig. 3 right). In contrast, our MTV framework faithfully generates audio-sync videos with cinematic quality.

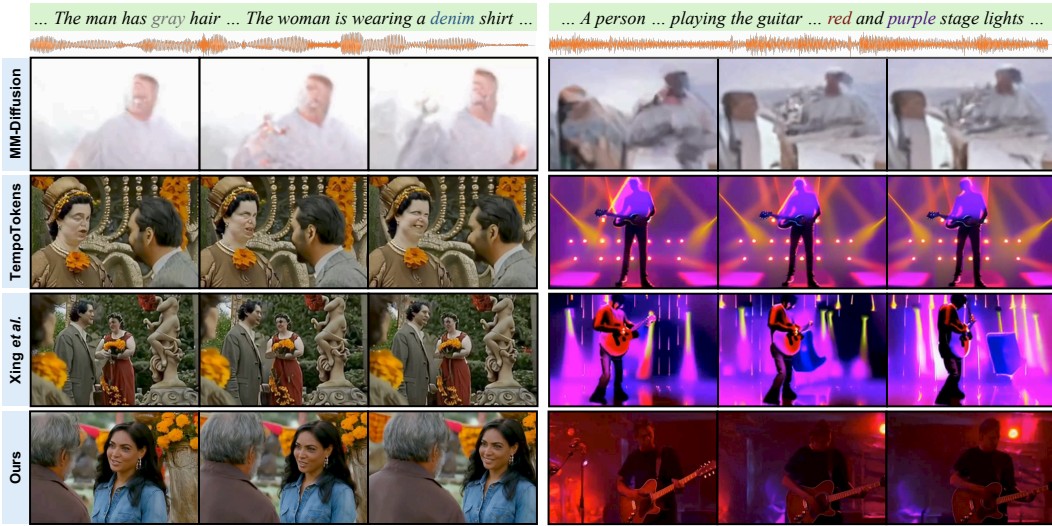

Figure 3: Visual comparison results with state-of-the-art methods for audio-sync video generation.

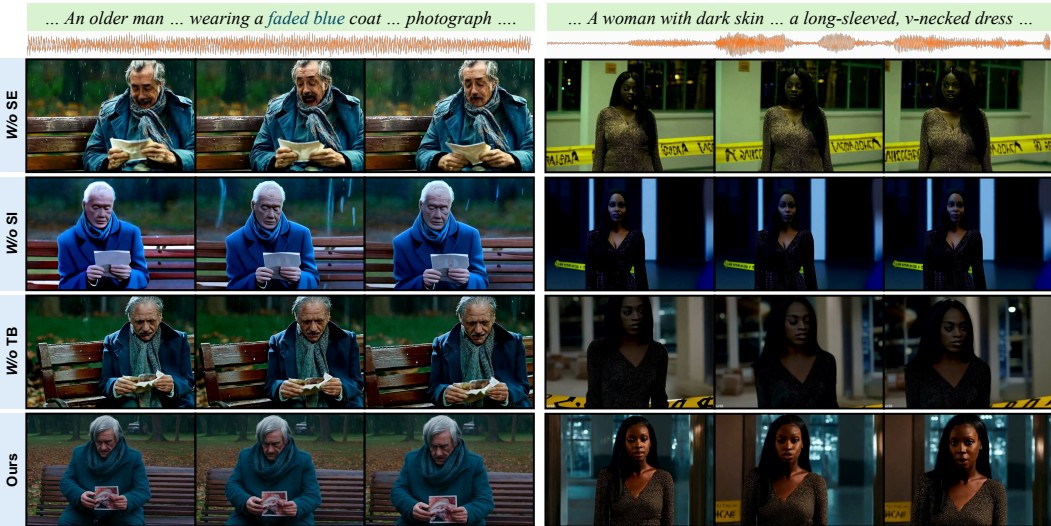

Figure 4: Ablation study results of different MST-ControlNet variants.

## 5.2 Ablation Study

To evaluate the effectiveness of key components within MST-ControlNet, we conduct ablation studies against three baseline configurations, as shown in Fig. 4 and Tab. 2.

**W/o SE (Separate Extraction).** We extract all features from demixed audio tracks using interval interaction blocks. This prevents music features from shaping the overall aesthetic presentation, leading to reduced visual mood (Fig. 4 left, degraded FVD and Temp-C).

**W/o SI (Separate Injection).** We extract features from demixed audio tracks by their respective encoders. These features are then concatenated and injected into the denoising network via a shared cross-attention. This reduces conditional consistency (Fig. 4 left, decreased Text-C and Audio-C).

**W/o TB (Training Backbone).** We freeze all weights of DiT backbone and only train our proposed MST-ControlNet to preserve more generative priors. This impairs the specific feature synchronization, especially the lip motion synchronization (Fig. 4 right, reduced Sync-C and Sync-D).

Table 3: User study results. Ours (MTV) clearly produces a higher score than state-of-the-art methods.

| Subjective criteria | MM-Diffusion [5] | TempoTokens [1] | Xing *et al.* [3] | Ours (MTV) |
|---|---|---|---|---|
| Semantic consistency | 0.96% | 13.60% | 11.28% | **74.16%** |
| Motion fluency | 0.64% | 8.96% | 12.56% | **77.84%** |
| Overall preference | 0.72% | 12.00% | 12.40% | **74.88%** |

Table 4: Quantitative experiment results with alternative pre-trained components.

| Method | FVD ↓ | Temp-C (%) ↑ | Text-C (%) ↑ | Audio-C (%) ↑ | Sync-C ↑ | Sync-D ↓ |
|---|---|---|---|---|---|---|
| CogVideoX+Wav2Vec | 626.06 | 95.40 | 26.55 | 26.22 | **3.17** | **9.43** |
| CogVideoX+Beats | 598.53 | 95.91 | 26.25 | 25.28 | 3.02 | 9.52 |
| Wan14B+Wav2Vec | **353.61** | **96.36** | **27.23** | **26.49** | 3.08 | 9.56 |

## 5.3 User Study

To better evaluate our method from a human perception perspective, we conduct three subjective user study experiments in Tab. 3. We present videos generated by our method and all baselines to participants and ask them to choose the best one based on the following criteria: *(i)* **Semantic consistency.** How well the video content aligns with the text description. *(ii)* **Motion fluency.** The realism and temporal coherence of the motion. *(iii)* **Overall preference.** How good the holistic quality of the video is. For each study, we randomly select 50 text descriptions from the test set, and the evaluations are conducted by 25 volunteers. The table below shows the percentage of times each method is chosen as the winner. Our method is consistently favored by human observers and has achieved the highest scores across all three subjective criteria.

## 5.4 Analysis of Pre-trained Components

We evaluate the robustness of our proposed method by integrating it with alternative pre-trained components. Specifically, we test replacing the audio encoder (Wav2Vec/BEATs) and the video backbone (CogVideoX/Wan14B) in Tab. 4.

**BEATs.** Since Wav2Vec [58] is a common setting for speech encoding (*e.g.*, Hallo3 [7]), this baseline only replaces it with BEATs [65] for both the effects and music tracks. As shown in Tab. 4, this baseline achieves comparable (or slightly better) video-related metrics (*i.e.*, FVD and Temp-C) but shows a slight degradation on audio-related metrics (*i.e.*, Audio-C, Sync-C, and Sync-D), suggesting that our current choice of Wav2Vec [58] is a robust and effective one for this task.

**Wan14B.** Since Wan14B [21] shares a similar DiT-based structure with CogVideoX [10], we can integrate our proposed MST-ControlNet into it without architectural changes. Specifically, our interval feature injection and holistic style injection modules are added after each text cross-attention layer. The quantitative results below show this baseline achieves better performance on video- and text-related metrics (*i.e.*, FVD, Temp-C, and Text-C) due to the stronger capabilities of the Wan14B [21], while achieving comparable performance on all audio-related metrics (*i.e.*, Audio-C, Sync-C, and Sync-D).

## 5.5 Application

As presented in Fig. 5, our model support four typical scenarios: *(i)* By integrating text-to-video generative priors and learned audio-visual synchronized capabilities, our model can create vivid virtual characters. *(ii)* Given user-provided images and taking them as arbitrary keyframes, our model can drive the image according to the given audios. *(iii)* Although our model generates video segments of 49 frames, it can achieve long video generation by using the generated frame to initialize the next segment. *(iv)* Following training-free approaches [66], our model can generate scene transitions guided by providing time-varying text descriptions.

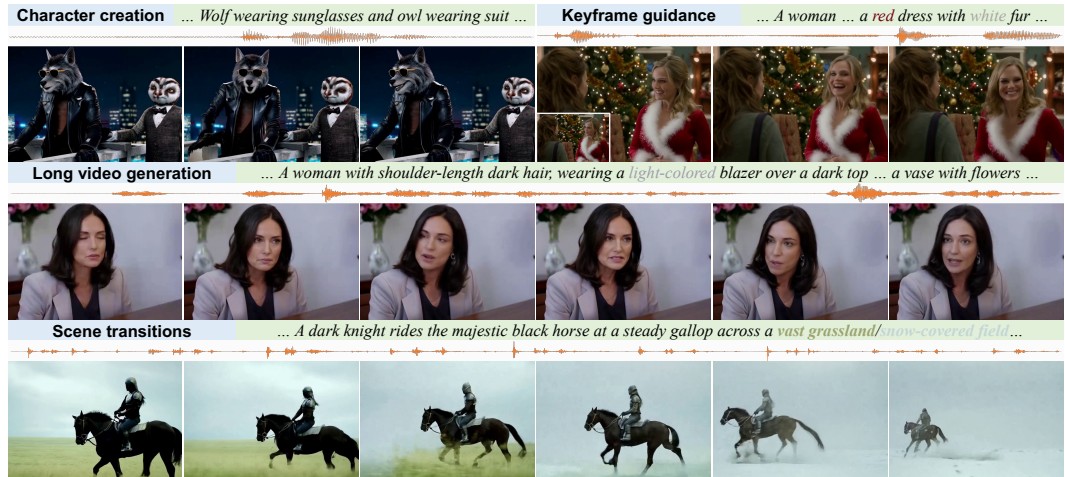

Figure 5: Examples of versatile application scenarios for our proposed MTV framework.

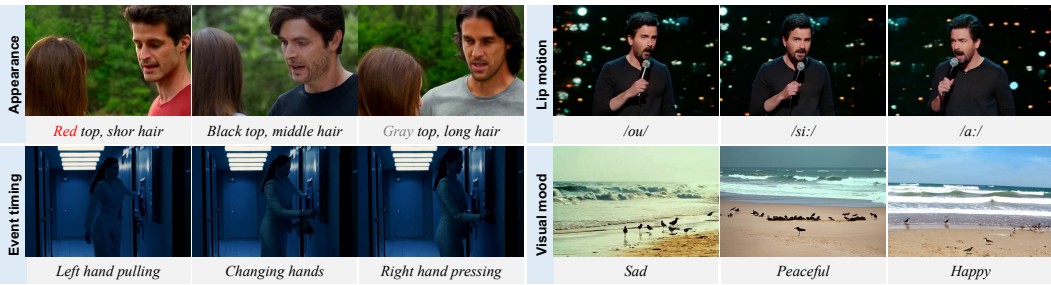

Figure 6: Examples of controllability study for text descriptions and demixed audios.

## 5.6 Controllability

As shown in Fig. 6, leveraging control from both text descriptions and the three demixed audio tracks (*i.e.*, speech, effects, music), our model can offer controllability across following four key aspects: *(i)* Modifying the text descriptions while keeping all audio tracks fixed allows the visual scene appearance to be edited without affecting the audio synchronization. *(ii)* Given a demixed speech track, the model enables precise control over the synchronized lip motion of the generated character. *(iii)* Similarly, with a demixed effects track, the model accurately synchronizes event timing with the sound effects. *(iv)* By changing the demixed music track, the model creates different visual moods for the generated video.

## 6 Conclusion

In this work, we presented MTV, a versatile framework for audio-sync video generation. MTV leverages generative priors from pretrained text-to-video models [10] and is trained on our contributed DEMIX dataset that provides sufficient cinematic videos with demixed audio tracks. Equipped with our proposed MST-ControlNet, MTV is able to independently control lip motion, event timing, and visual mood. Combined with a multi-stage training strategy for effective learning of complex audio-visual relationships, MTV achieves state-of-the-art performance across six evaluation metrics.

**Limitation.** Although our approach demonstrates the potential of using demixed audio tracks for precise video control, it is fundamentally limited by the scope of categories provided by upstream audio demixing techniques [53, 54]. We believe the capabilities of audio-sync video generation methods will further progress with advancements in audio demixing methods.

**Acknowledgement.** This work is supported by National Natural Science Foundation of China (Grant No. 62136001). We thank all the insightful reviewers for the helpful suggestions, and the colleagues at Beijing Academy of Artificial Intelligence for their support throughout this project.

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
