# OpenReview forum: "Audio-Sync Video Generation with Multi-Stream Temporal Control"
_NeurIPS.cc/2025/Conference — NeurIPS 2025 poster_

### Official Review · Reviewer_1iMS · 2025-06-24

**Clarity:** 4
**Significance:** 3
**Originality:** 3
**Rating:** 5
**Confidence:** 4

**Summary:**

This paper presents a new method to adapt video generation models for audio-to-video synthesis. It assumes that conditional audio is given as three separate tracks: speech, sound effects, and music. Under this setting, the proposed model extracts features from the three tracks via Multi-stream Temporal ControlNet and feeds them into the pretrained video generation model through additional cross-attention and feature modulation depending on the type of track. To train and evaluate the model, the authors created a new dataset comprising video, text, and demixed audio tracks. In the experiments, the proposed model outperforms prior audio-to-video models in terms of both video quality and audio-visual alignment.

**Questions:**

- Did the authors conduct any filtering process regarding license or copyright of the video when constructing the dataset?

**Ethical Concerns:**

["Major Concern: Data privacy, copyright, and consent"]

**Final Justification:**

I maintain my rating to be 5.

I raised a concern on the copyright issue of the dataset. This was addressed in the rebuttal and resolved by the ethics review. Thus, I do not have any reason to downgrade my rating.

**Limitations:**

The limitations of the proposed method are briefly discussed in Section 6.

**Paper Formatting Concerns:**

I have no concerns regarding paper formatting.

**Quality:**

4

**Strengths And Weaknesses:**

## Strengths

- The idea of using demixed audio for audio-to-video synthesis is interesting and convincing.
  - Explicitly handling such separate audios should enable fine-grained controls to the video generation process, because how conditioning audio impacts on the generated videos should vary depending on the type of the audio.
- The dataset construction process seems to be thorough, and the release of this dataset would benefit a lot the community in the field of audio-visual generation.
- The proposed model empirically outperforms prior audio-to-video models significantly. I have checked the supplementary video and the performance gain is also obvious in perceptual quality.
- The manuscript is well-written and easy to follow.

## Weaknesses

- I have a concern regarding license or copyright issues on the film-related videos mentioned in line 111.
  - This is because the dataset construction process shown in Section 3 seems not to contain a filtering process to eliminate this issue.
- I do not find crucial weaknesses regarding the design of the proposed method and the experiments. Shown below are just a few suggestions to strengthen the significance of this study rather than weaknesses. It is completely fine to ignore them during the rebuttal phase.
  - In the proposed model, the extracted features \tilde{f} are commonly used for every layer of DiT, and this design is essentially different from the standard setting of ControlNet for DiT adopted in popular models such as Stable Diffusion 3.5 and PixArt-delta, where the features are extracted in a layer-wise manner. I understand that the standard design cannot be directly applied to the case in this study due to the domain gap between video and conditional audio, but it would be worth trying layer-adaptive feature extraction also in the proposed model. I personally suspect that the above-mentioned design could contribute the trade-off mentioned in Section A.2.
  - Regarding the generalizability, it would be great if the proposed method also works with more recent video diffusion models such as Wan, but I understand that trying this could require prohibitively large computational cost for training.

---

> ### Author Rebuttal · Authors · 2025-07-30
>
> We thank the reviewers for their insightful and constructive feedback.
>
> In accordance with the rebuttal policy, we are unable to include new qualitative results (e.g., generated videos) or external links. We assure the reviewers that, should the paper be accepted, the camera-ready version will be thoroughly updated to include:
>
> 1. All additional experiments and qualitative results are discussed in this rebuttal.
>
> 2. A public link to an open-source demo showcasing our MTV framework as a practical video generation tool.
>
> We now address the specific points raised by the reviewers.
>
>
>
> **W1: Copyright**
>
> We thank the reviewer for this important question. We take data sourcing and copyright considerations very seriously and have followed established academic practices.
>
> 1. Data sources:
>
> As detailed in our paper (Sec. 3, Lines 109-112), our DEMIX dataset is constructed from two main sources:
>
> Existing Public Datasets: We use several well-established benchmarks, including CelebV-HQ, MovieBench, Condensed Movies, and Short-Films 20K.
>
> Publicly Available Videos: We also collect data from YouTube, mirroring the widely-used academic datasets like VGGSound.
>
> 2. Release justification:
>
> For videos from existing datasets, we will follow their original protocols by providing the video IDs and other necessary metadata for users to retrieve the data from the original sources.
>
> For videos we collected from YouTube, we will provide a list of the corresponding URLs and timestamps.
>
> For our contributions, we will directly provide our generated annotations, including text descriptions and the demixed audio tracks.
>
> We do not redistribute the original video/audio content. The dataset is compiled strictly for academic research.
>
> 3. Licensing:
>
> To ensure clarity, we will adopt a dual-licensing approach similar to VGGSound:
>
> Our code, annotations, and data-processing scripts will be released under the Apache 2.0 License.
>
> We will explicitly state that the copyright of the underlying video and audio content remains with their original owners.
>
>
> **W2: Layer-adaptive feature extraction**
>
> We sincerely thank the reviewer for this insightful suggestion and for proposing an interesting alternative design.
>
> Our current design (audio features are injected into all DiT blocks) follows the established practice in state-of-the-art talking face generation models (e.g., FantasyTalking). Since this approach has been proven effective and robust, we adopt it as our primary design.
>
> To explore the reviewer's suggestion, we are implementing a layer-wise approach. We find that directly adapting the standard DiT blocks of CogVideoX results in out-of-memory errors, even using eight A800 80G GPUs. Therefore, we are proceeding by modifying our own MTSControlNet to perform layer-wise injection, where specific blocks of audio features are injected into corresponding blocks of the DiT backbone.
>
> Due to the time constraints of the rebuttal period, this experiment is currently in progress. We will post the results and our analysis as soon as they are available.
>
> **W3: Adaptability to additional T2V backbone**
>
> Our proposed MSTControlNet can be seamlessly integrated into various T2V backbones (e.g., OpenSora and Wan) without architectural changes, as they share a similar DiT-based structure. Due to time limitations, we only implement and test the integration with Wan14B. Specifically, our interval feature injection and holistic style injection modules are added after each text cross-attention layer.
>
> For inference, MTV (Wan14B backbone) requires 180s to generate an 81-frame, audio-synchronized video at a 480P resolution on 8 NVIDIA A100 GPUs. We report the new quantitative results below.
>
> | Method | FVD ↓ | Temp-C (%) ↑ | Text-C (%) ↑ | Audio-C (%) ↑ | Sync-C ↑ | Sync-D ↓ |
> | :--- | :---: | :---: | :---: | :---: | :---: | :---: |
> | CogVideoX | 626.06 | 95.40 | 26.55 | 26.22 | **3.17** | **9.43** |
> | Wan14B | **353.61** | **96.36** | **27.23** | **26.49** | 3.08 | 9.56 |
>
> **Q1: Copyright**
>
> See W1 please.

---

> > ### Comment · Reviewer_1iMS · 2025-08-02
> > **Thanks for the response**
> >
> > Thanks for the clarification on copyright and also for providing additional empirical results.
> >
> > I have no further questions for the authors. At this moment, I would like to maintain my rating and proceed to the discussion with the other reviewers.

---

> > > ### Author Response · Authors · 2025-08-02
> > >
> > > We thank the reviewer for their positive feedback and support of our work.

---

> ### Author Response · Authors · 2025-08-07
> **Experiment results for layer-adaptive injection**
>
> Following up on our previous response, we have now completed the experiments regarding your insightful suggestion on layer-adaptive feature extraction. We thank you again for motivating this valuable ablation study.
>
> As we state previously, directly adapting standard DiT blocks results in out-of-memory errors. Therefore, we implement the suggestion by modifying our MST-ControlNet to inject features from its different layers into the corresponding blocks of the DiT backbone. Since the CogVideoX backbone has 42 DiT blocks, we test configurations where the number of MST-ControlNet blocks (and thus the number of injection points) is set to N=4, 21, and 42.
>
> As result presented below, these metrics reveal a complex trade-off:
>
> * For audio-related metrics (Audio-C and Sync-C/D), the metrics generally improve as more blocks are used, indicating stronger audio synchronization.
>
> * For the temporal consistency of generated videos (Temp-C), the metric degrads as the number of blocks increases.
>
> * For overall quality and text alignment (FVD and Text-C), the metrics fluctuate without a clear trend.
>
>
> | Method   | FVD ↓  | Temp-C (%) ↑ | Text-C (%) ↑ | Audio-C (%) ↑ | Sync-C ↑ | Sync-D ↓ |
> | :------- | :----: | :----------: | :----------: | :-----------: | :------: | :------: |
> | block=4  | 630.64 |   95.19   |   26.51    |    24.26    |   1.78   |  10.50   |
> | block=21 | 639.98 |   94.71    |   26.33    |   23.61    |   2.43   |   9.54   |
> | block=42 | 633.50 |   94.40    |  26.40    |    23.03    |   2.75   |   9.48   |
> | Ours (MTV) | **626.06** |   **95.40**    |   **26.55**    |    **26.22**    |   **3.17**   |   **9.43**   |
>
>
> As a result, increasing the number of blocks in a layer-wise injection approach improves audio synchronization, but this comes at the cost of video fluency. It appears that forcing the model to adapt to audio features at too many layers overly prioritizes synchronization at the expense of the video's natural temporal flow. We hypothesize this is because audio features from earlier MST-ControlNet layers are less refined, which means the backbone's early layers do not receive sufficient high-quality audio cues.
>
> Notably, although we test these different layer-wise configurations, our original proposed architecture still achieves the best overall performance. This suggests that for our task, layer-wise injection may not be the optimal approach.

---

> > ### Comment · Reviewer_1iMS · 2025-08-08
> >
> > Thanks for providing additional results.
> >
> > The performance trend across the number of blocks is interesting, and the authors' discussion sounds reasonable.

---

### Official Review · Reviewer_3d1R · 2025-07-03

**Clarity:** 3
**Significance:** 3
**Originality:** 2
**Rating:** 4
**Confidence:** 4

**Summary:**

This paper introduces MTV for generating video synchronized with audio input by explicitly separating audio into three tracks: speech, effects, and music. This separation allows precise control over lip motion, event timing, and visual mood.
In addition, the authors introduce DEMIX, a new large-scale dataset composed of cinematic video clips paired with demixed audio.
The paper also proposes a new component, the Multi-Stream Temporal ControlNet (MST-ControlNet), designed to handle these separate audio streams and integrate their signals into a text-to-video diffusion model. The paper reports state-of-the-art results across several metrics on audio-sync video generation, including video quality, audio-video synchronization, and consistency with text descriptions. The authors also provide extensive ablations.

**Questions:**

1. How are the other 2 baselines (TempoTokens and Xing et al.) evaluated? Are they retrained on the DEMIX dataset?
2. Could the authors provide more details on the fine-tuning of MM-Diffusion? Does it follow the same multi-stage training process on all five subsets of DEMIX?
3. Could the authors provide examples where the audio demixing is not perfect and show how this negatively impacts the final video?
4. Could the authors provide examples with multiple people and complicated scenarios for text descriptions?

**Ethical Concerns:**

["NO or VERY MINOR ethics concerns only"]

**Final Justification:**

I have raised my score for this paper from 2 to 4.

My initial score (2: Reject) was due to significant concerns about comparison fairness, the novelty of the contributions, the model's reproducibility, and the lack of experiments to show the model's robustness.

The authors' rebuttal was convincing. They addressed every major issue by providing extensive new results, which included: (1) Experiments on different source separation levels. (2) Evaluating on public benchmarks to show generalization. (3) Conducting new ablations with a different T2V backbone (Wan14B) and different pre-existing models. (4) Adding a human evaluation study. (5) Experiments on comparison fairness.

Also, my follow-up questions and concerns (training schedule, data augmentation, and training on noisy separations) were clearly addressed.

The author also claimed that all source code and pre-trained models will be released.

Based on these, I am now much more positive about this work, and therefore raised my score from 2 to 4.

**Limitations:**

Yes.

**Paper Formatting Concerns:**

No.

**Quality:**

3

**Strengths And Weaknesses:**

Strengths:
1. This paper targets a meaningful and useful gap in video generation and audio-video synchronization, especially in the field of cinematic video.
2. This paper builds a new dataset, DEMIX, with a large-scale and careful filtering, which is a solid contribution to the community.
3. The paper provides remarkable quantitative and qualitative results over three baseline models, indicating the effectiveness of the proposed method.
4. Paper writing is clear and easy to understand.

Weaknesses:
1. The proposed model is heavily dependent on a list of existing models. For example:
(1) Audio track separation has a significant impact on the model's overall performance (Tab 2 ablation study), while it is heavily bottlenecked by the demixing tools (MVSEP, Spleeter). The authors didn't provide failure cases when audio is noisy and is not cleanly separable, which is quite common in real-world scenarios.
(2) The proposed model applies CogVideoX as base model, wav2vec for audio encoding, and T5 for text embedding. I believe the novelty of architecture mainly lies in module assembling. Given that, it is unclear how much of the performance gain comes from the proposed method instead of pretrained models. More ablations on choosing different pre-existing models are suggested.
(3) Given the complicated pipeline that includes multiple external models, I have concerns on model's reproducibility.
2. From my understanding, the text template is fixed (Sec 4.3), which lacks generalizability when the scenario is complicated to describe. There lacks such examples (e.g, 3 or more people) discussed in the paper. There lack of discussion on limitations of text descriptions, which I believe has a high impact on the overall performance and control quality.
3. Concerns about comparison fairness. It is unclear to me if the other 2 baselines (TempoTokens and Xing et al.) are retrained on the DEMIX dataset or not. Without consistent training and evaluation approaches, the fairness of performance comparisons is questionable. Experiments on public standard benchmark are suggested.

---

> ### Author Rebuttal · Authors · 2025-07-30
>
> We thank the reviewers for their insightful and constructive feedback.
>
> In accordance with the rebuttal policy, we are unable to include new qualitative results (e.g., generated videos) or external links. We assure the reviewers that, should the paper be accepted, the camera-ready version will be thoroughly updated to include:
>
> 1. All additional experiments and qualitative results are discussed in this rebuttal.
>
> 2. A public link to an open-source demo showcasing our MTV framework as a practical video generation tool.
>
> We now address the specific points raised by the reviewers.
>
> ## ***Weakness***
>
> **W1-1: Unclear source separation**
>
> We agree that audio demixing tools are not always perfect. However, we would like to clarify that all of our reported results are generated from these imperfectly separated audio tracks, not from idealized clean inputs. As shown in Tab. 2 and Fig. 3, our framework achieves state-of-the-art performance across six standard metrics, directly demonstrating its inherent robustness in real-world scenarios. This stems from our demixing filtering strategy (Lines 117-122). We explicitly filter out extremely poor-quality samples during training, which enables our model to learn precise audio-visual synchronization,  while the remaining unclear separations serve as a form of data augmentation that enhances robustness.
>
> We additionally create three quality-based subsets (Level 1 being the highest quality, Level 3 the lowest), with each subset containing 20 samples. The quality level for each sample is determined by a consensus rating from 5 human volunteers. We also evaluate a "Random" subset, which is randomly sampled from the entire test set for baseline comparison.
>
> As the results below show, our model is robust to the quality of the demixed audio. Its performance on metrics like FVD and Text-C shows only minor fluctuations across the quality levels, while audio-visual correlation metrics (Audio-C and Sync-C) are even slightly better on lower-quality input. This may suggest that our model effectively focuses on audio cues that persist even when the demixed audio is not perfectly clean. Notably,  with any-level quality inputs, our model remains superior to all baselines evaluated on the Random subset. Therefore, our multi-stream approach provides robust gains overall.
>
> | Demix quality | FVD ↓ | Temp-C (%) ↑ | Text-C (%) ↑ | Audio-C (%) ↑ | Sync-C ↑ | Sync-D ↓ |
> | :--- | :---: | :---: | :---: | :---: | :---: | :---: |
> | MTV (Level1) | **617.23** | 95.14 | **27.03** | 24.81 | 2.88  | **8.99** |
> | MTV (Level2) | 665.18 | 95.60 | 26.66 | 26.01 | 3.13 | 9.29 |
> | MTV (Level3) | 638.92 | **95.88** | 26.52 | **26.87**| **3.46** | 9.41 |
> | MTV (Random) |  626.06 | 95.40 | 26.55 | 26.22 | 3.17 | 9.43 |
> | MM-Diffusion (Random) | 879.77 | 94.15 | 15.61 | 5.43 | 1.53 | 11.21 |
> | TempoTokens (Random) | 795.88 | 93.13 | 24.68 | 6.71 | 1.45 | 10.48 |
> | Xing et al. (Random) | 805.23 | 93.30 | 24.51 | 7.30 | 1.55 | 10.50 |
>
> Finally, to completely bypass this separation challenge, we further developed a demo. It uses Qwen3 to interpret user prompts into audio descriptions, synthesizes them into perfectly clean audio with ElevenLabs, and then generates a synchronized video with our MTV framework. This presents a complete text-to-video pipeline to avoid potential source separation errors entirely.
>
> **W1-2: Different pre-existing models**
>
> We create two new baselines by swapping key pre-trained components to evaluate how much of the performance gain comes from our proposed method.
>
> 1. Wan14B with umT5. Since Wan14B shares a similar DiT-based structure with CogVideoX, we can integrate our proposed MTSControlNet into it without architectural changes. Specifically, our interval feature injection and holistic style injection modules are added after each text cross-attention layer. The quantitative results below show this baseline achieves better performance on video and text metrics (FVD, Temp-C, Text-C) due to the stronger capabilities of the Wan14B and umT5 models, while achieving comparable performance on all audio-related metrics (Audio-C, Sync-C, and Sync-D). This demonstrates that our MTSControlNet maintains its high performance on audio alignment and lip synchronization, independent of the specific T2V backbone.
>
> 2. Beats. Since Wav2Vec is a common setting for speech encoding (e.g., Hallo3), this baseline only replaces Wav2Vec with Beats for both the effects and music tracks. As shown in the table below, this baseline achieves overall comparable performance, suggesting that our current choice of Wav2Vec is a robust and effective one for this task. This finding is consistent with previous work in Visual-Animation [1] (see its Appendix 6.7).
>
> | Method | FVD ↓ | Temp-C (%) ↑ | Text-C (%) ↑ | Audio-C (%) ↑ | Sync-C ↑ | Sync-D ↓ |
> | :--- | :---: | :---: | :---: | :---: | :---: | :---: |
> | CogVideoX+T5+Wav2Vec  |626.06 |95.40 | 26.55 | 26.22 | **3.17** | **9.43** |
> | Wan+umT5+Wav2Vec |**353.61** | **96.36** | **27.23** | **26.49** | 3.08 | 9.56 |
> | CogVideoX+T5+Beats | 598.53 | 95.91 | 26.25 | 25.28 | 3.02 | 9.52 |
>
> Notably, pre-trained T2V generators are typically tightly coupled with their specific text encoders. Therefore, replacing only the video generator or its text encoder separately would require fine-tuning the entire backbone, which requires prohibitive computational resources and is not feasible within the rebuttal period.
>
> [1] Audio-Synchronized Visual Animation. ECCV, 2024.
>
> **W1-3: Reproducibility**
>
> We will release the source code and pre-trained models upon acceptance.  Should the reviewer have questions about any specific implementation details during the rebuttal period, we are glad to post the corresponding code directly in the rebuttal response.
>
> **W2: Complicated scenario**
>
> The primary purpose of our text template is not to describe every detail of the scene, but to effectively specify the active speakers, their appearance, and the scene in which they are present. Based on our empirical observation that most one-shot scenarios have fewer than three active speakers, we initially set the upper bound of active speakers to two.
>
> While this number can be increased to three or more with fine-tuning, the true bottleneck for complex scenes lies in the token limit of the text encoder (i.e., T5's 512-token limit). This inherent constraint of the T2V backbone prevents users from providing long descriptions for every single person, regardless of whether a structured template is used. As a result, for scenarios with complex backgrounds (e.g., a crowd of people), these details can be concisely described and included in the "scene description" portion of our template (e.g., in a bustling city square full of people). We will include such cases in the final version.
>
> We further conduct an additional ablation study by fine-tuning our model using only free-form text descriptions instead of our structured template. As the results below show, the video quality is degraded, demonstrating the effectiveness of our text template design.
>
> | Method | FVD ↓ | Temp-C (%) ↑ | Text-C (%) ↑ | Audio-C (%) ↑ | Sync-C ↑ | Sync-D ↓ |
> | :--- | :---: | :---: | :---: | :---: | :---: | :---: |
> | Free form | 638.89 | 95.42 | 26.45 | 25.17 | 2.92 | 9.45 |
> | Text template | **626.06** | **95.40** | **26.55** | **26.22** | **3.17** | **9.43** |
>
> **W3: Comparison fairness**
>
> In our initial submission, we evaluate TempoTokens and Xing et al. using their publicly released weights. To make a fair comparison, we fine-tune all baseline methods (MM-Diffusion, TempoTokens, and Xing et al.) on our DEMIX dataset using their official training schedules, and re-evaluate them on our DEMIX dataset.  The updated results are presented below. Our proposed MTV framework still maintains a significant lead, although TempoTokens and Xing et al. show notable improvement in lip synchronization after fine-tuning.
>
> | Method | FVD ↓ | Temp-C (%) ↑ | Text-C (%) ↑ | Audio-C (%) ↑ | Sync-C ↑ | Sync-D ↓ |
> | :--- | :---: | :---: | :---: | :---: | :---: | :---: |
> | MM-Diffusion |879.77|94.15|15.61 | 5.43 | 1.53 | 11.21 |
> | TempoTokens | 883.35 | 92.61 | 24.21 | 6.06 | 1.85 | 10.26 |
> | Xing et al. | 847.85 | 92.79 | 24.69 | 6.11 | 1.78 | 10.03 |
> | Ours (MTV) |  **626.06** | **95.40** | **26.55** | **26.22** | **3.17** | **9.43** |
>
> We further follow TempoTokens to conduct evaluations on both the Landscape and AudioSet-Drum datasets.  As shown in the tables below, our MTV framework still achieves significantly better performance.  Since neither dataset includes human talking, the lip synchronization metrics (Sync-C and Sync-D) are not applicable for these evaluations.
>
> #### **Comparison on Landscape datasets**
>
> | Method | FVD ↓ | Temp-C (%) ↑ | Text-C (%) ↑ | Audio-C (%) ↑ |
> | :--- | :---: | :---: | :---: | :---: |
> | MM-Diffusion | 807.65 | 94.74 |14.66| 16.59 |
> | TempoTokens | 797.33 | 94.67 | 21.73 | 18.86 |
> | Xing et al. | 838.03 | 94.71 | 21.04 | 18.70 |
> | Ours (MTV) | **697.51** | **96.98** | **25.35** | **23.37** |
>
> #### **Comparison on AudioSet-Drum dataset**
>
> | Method | FVD ↓ | Temp-C (%) ↑ | Text-C (%) ↑ | Audio-C (%) ↑ |
> | :--- | :---: | :---: | :---: | :---: |
> | MM-Diffusion | 1520.09 | 94.59 | 14.90 | 14.11 |
> | TempoTokens  | 1512.97 | 94.28 | 23.18 | 15.59 |
> | Xing et al. | 1589.46 | 94.49 | 23.73 | 17.84 |
> | Ours (MTV) | **1511.53** | **97.50** | **25.62** | **39.61** |
>
> **Q1: Retrained baselines**
>
> See W3 please.
>
> **Q2: Finetuning MM-Diffusion**
>
> For fine-tuning MM-Diffusion, we strictly follow the official training schedule: We first train the coupled U-Net, and subsequently train its super-resolution model. Both stages are performed on our full DEMIX dataset.
>
> We do not apply our training schedule to comparison methods. As this is the contribution of our MTV framework, applying it to other methods would prevent a fair evaluation of our improvement.
>
> **Q3: Imperft audio demixing**
>
> See W1-1 please.
>
> **Q4: Complicated scenarios**
>
> See W2 please.

---

> ### Author Response · Authors · 2025-08-06
>
> Dear Reviewer, as the discussion period is ending soon, we would be grateful for any feedback on our rebuttal and are happy to answer any further questions.

---

> > ### Comment · Reviewer_3d1R · 2025-08-06
> >
> > Thank you for your comprehensive rebuttal and the significant effort you've invested in conducting additional experiments to address the concerns raised by me and the other reviewers.
> >
> > Your new results, including the retraining of all baselines on DEMIX, the evaluation on public benchmarks (Landscape, AudioSet-Drum), and the ablation studies on the T2V backbone and text template, have substantially strengthened the paper. I particularly appreciate the clarification on comparison fairness, which was one of my primary concerns.
> >
> > These additions have addressed many of my initial reservations. However, after carefully reading all the rebuttals and discussing them with other reviewers, a few points of confusion remain:
> >
> > 1. In your rebuttal, you demonstrate that MST-ControlNet can be integrated into a different backbone (Wan14B), and you also show that your multi-stage training strategy is critical for performance (rebuttal to Reviewer 453Q). These are both strong points. Also, my understanding is that your multi-stage training relies on the five specific subsets of DEMIX.
> > *Is this training schedule universal, or is it tailored to the CogVideoX backbone and the DEMIX dataset?  Let's say, if one were to adapt MTV to a new model like Wan14B, would the multi-stage training process, including the dataset subsets, need to be redesigned? Clarifying the "seamlessness" of adapting the full framework (not just the ControlNet component) would be helpful.*
> >
> > 2. In your rebuttal, you make an interesting point that imperfectly demixed audio samples serve as a form of data augmentation, enhancing the model's robustness. *Could you explain more about this? Is there a risk that by training on noisy separations, the model will learn to ignore subtle but important audio cues and instead focus on more obvious, less ambiguous sounds? How does the model learn to distinguish between meaningful audio events in a slightly noisy track versus pure demixing artifacts?*
> >
> > I am now much more positive about this work based on the additional results you provided in the rebuttal, and I believe it has the potential to be a strong contribution. If you can provide clear and convincing answers to the above remaining questions, I will be happy to raise my score.

---

> > > ### Author Response · Authors · 2025-08-06
> > >
> > > Thank you for thoughtful and positive feedback on our work. We appreciate the opportunity to provide further clarification. We address each of your remaining points below and welcome any further discussion.
> > >
> > > **Q1: Training schedule**
> > >
> > > We believe this training schedule is universal. The guiding idea is to train the model progressively, starting from concrete and localized controls (e.g., lip motion) and advancing towards more abstract and global controls (e.g., visual mood). This approach is not tied to any specific backbone.
> > >
> > > To be precise, when we integrate our MST-ControlNet into the Wan14B backbone, we use the exact same multi-stage training schedule and dataset subsets. Specifically, we fine-tune the model on each subset for approximately 2K steps, which takes about 22 hours on 24 NVIDIA A800 GPUs. Thanks to the stronger generative priors from Wan14B, the model converges much faster. This allows the finetuning process to focus primarily on learning the audio-visual alignment, rather than foundational video quality.
> > >
> > > As demonstrated in our previous response (Ablation "Mix" for Reviewer 453Q, L2), training the model on the full DEMIX dataset without this multi-stage strategy will result in a significant degradation in lip synchronization performance. This confirms the critical importance of this approach.
> > >
> > > To adapt our framework to a new dataset, we require the data to be divided into conceptually similar subsets (i.e., basic face, single character, multiple characters, sound event, and visual mood) to maintain the principle of "learning from concrete towards abstract." To facilitate this, we will release our data processing scripts to assist other researchers in structuring their own datasets accordingly.
> > >
> > > **Q2-1: Data augmentation**
> > >
> > > We do not train on all imperfect samples. As detailed in the paper (Lines 117-128) , we explicitly filter out and discard clips where the separation quality is extremely low. This ensures that the primary audio signal (e.g., speech) typically remains dominant and clearly correlated with the visual event in our training data. The residual noise is kept at a low level, which prevents the model from learning to ignore the main signal.
> > >
> > > Furthermore, even when not perfectly clean, the demixed tracks retain the majority of the original audio cues with high fidelity. As our response to Reviewer uen2 (Q4) , we find that by directly summing the three demixed tracks and comparing the result to the original audio, that mean L1 distance is just 0.000863, demonstrating minimal information loss.
> > >
> > > **Q2-2: Audio cues**
> > >
> > > Our model learns to distinguish between meaningful signals and demixing artifacts through statistical correlation on a large scale. Specifically, a meaningful audio event (e.g., a specific word, a door slam) will have a strong and consistent correlation with its corresponding visual event across the thousands of samples in the DEMIX dataset. In contrast, demixing artifacts are typically inconsistent and lack a predictable relationship with the primary visual events. The network naturally learns to assign high importance to these reliable patterns (the signal) and ignore unreliable ones (the noise).
> > >
> > > Architecturally, our model is designed to adaptively integrate cross-track cues. Our empirical observation is that the most critical cues for fine-grained synchronization lie in the speech and sound events. This motivates us to apply a self-attention mechanism within the interval stream across the features of both the speech and effect tracks, allowing the model to learn their interplay at each time interval.
> > >
> > > With these designs in place, the entire process is guided by the optimization objective. The model is trained to minimize the difference between the generated video and the ground truth. If the model ignores key audio cues, the resulting video (e.g., with incorrect lip movement) will have a significantly different distribution from the ground truth, leading to a large optimization penalty. This fundamentally forces the model to learn the audio-visual alignment effectively to achieve its goal.

---

> > > > ### Comment · Reviewer_3d1R · 2025-08-06
> > > >
> > > > Thank you for the clear and convincing follow-up. Your answers regarding the universality of the training schedule and the model's ability to learn from imperfect audio have fully resolved my remaining questions.
> > > >
> > > > Since all of my initial and follow-up concerns have been thoroughly addressed, I'm more positive about this work, and I have raised my score from 2 to 4. Please make sure all of these new changes are included in the final version of the paper.

---

> > > > > ### Author Response · Authors · 2025-08-07
> > > > >
> > > > > We sincerely thank the reviewer for their positive feedback and for raising their score.
> > > > >
> > > > > We will be sure to incorporate all the new results  in the final version.

---

### Official Review · Reviewer_uen2 · 2025-07-03

**Clarity:** 3
**Significance:** 3
**Originality:** 3
**Rating:** 4
**Confidence:** 4

**Summary:**

This paper presents MTV, a unified and controllable framework for audio-synchronized video generation. The framework is supported by a newly introduced dataset, DEMIX, which includes high-quality cinematic videos with corresponding audio tracks and is structured to support scalable training across varied video generation tasks.
The proposed approach demonstrates state-of-the-art performance in terms of video quality, text-video consistency, and audio-video alignment.

**Questions:**

1. Is it reasonable to use the same Wav2vec encoder to process all types of audio signals (speech, sound effects, and music)?
Wav2vec is primarily designed to extract semantic representations from speech. While this may be effective for speech, the authors should consider whether using alternative encoders for sound effects and music could yield better performance or more meaningful features.

2. In the proposed "multi-stream temporal ControlNet", I am particularly curious why speech and sound effects are mixed as input.
In clean talking-head scenarios, sound effects may interfere with accurate lip synchronization.
The authors should validate this design choice and provide a reasonable explanation for combining these audio streams.

3. How does the method handle multiple speakers? (Figure 1, second row). How are different speech signals distinguished and mapped to corresponding face regions? Would this setup lead to interference in visual information when speech overlaps or when there is temporal misalignment?

4. Is it sufficient to divide the input audio signal into only three components: speech, sound effects, and music?
The current categorization overlooks other potentially important audio modalities such as ambient sounds and reverb/spatial cues.

If the authors can provide clear and detailed responses to the concerns raised, I would be willing to reconsider and potentially raise my overall score.

**Ethical Concerns:**

["NO or VERY MINOR ethics concerns only"]

**Final Justification:**

Although the model design of this paper is somewhat blurred and lacks novelty, the authors' well-reasoned explanation clearly addresses these issues.

Besides, the choice of metrics requires careful consideration (including emotion/mood evaluation and multi-speaker AV-sync).
The evaluation of Emotional aligning is still a bit far-fetched by choosing FVD whether close to GT.

Overall, this paper is very meaningful for community, and I'm pleased to have improved my score to "4 Borderline Accept." More importantly, I hope the authors can public the source code to facilitate better community evaluation and development.

**Limitations:**

yes

**Quality:**

3

**Strengths And Weaknesses:**

Strengths

- The work is overall very clearly written and expressed.

- The author provides a new dataset DEMIX, and the corresponding view and experiment is supported

Weaknesses

- The embeddings are uniformly applied to all frames via global style injection to control visual emotion (Lines 56–57).
Is such global control reasonable? For long-sequence video modeling, mood distribution may not be uniform over time.
The authors are advised to provide a proper reference for the statement "Since visual mood typically covers the entire video clip" instead of relying on subjective assumptions.

- The "multi-stream temporal ControlNet" essentially injects audio information into the DiT via cross-attention. This approach appears overly simplistic and lacks novelty.

- The paper emphasizes the extraction of features from the music track to enhance visual mood. However, the experimental section does not include any objective evaluation metrics related to emotion or mood to support this claim.

---

> ### Author Rebuttal · Authors · 2025-07-30
>
> We thank the reviewers for their insightful and constructive feedback.
>
> In accordance with the rebuttal policy, we are unable to include new qualitative results (e.g., generated videos) or external links. We assure the reviewers that, should the paper be accepted, the camera-ready version will be thoroughly updated to include all additional experiments and qualitative results discussed in this rebuttal.
>
> We now address the specific points raised by the reviewers.
>
> **W1: Global style control**
>
> We agree that for long-sequence videos, mood distribution may not be uniform. However, we would like to clarify that our approach follows the common and well-justified practices of both affective computing and video generation.
>
> * Affective computing
>
> Our work focuses on generating video clips, typically under 10 seconds. In such durations, assuming a consistent visual mood is a standard approach. For example,  the LIRIS-ACCEDE [1] dataset (a general video dataset for affective content analysis) consists of video clips between 8-12 seconds, each annotated with a global Arousal-Valence value. The authors explicitly justify this choice, stating that using these durations "greatly minimizes the probability that annotations are a weighted average of consecutive emotions".
>
> Furthermore, this principle of applying a global label is also adopted in the highly influential DEAP dataset [2] for music videos, where even one-minute clips are assigned a single set of ratings (Valence, Arousal, Dominance, etc.) for the entire duration.
>
> * Video generation
>
> In our MTV framework, music features serve as a representation of the overall aesthetic, which is analogous to a global style for the generated video. Enforcing such global style consistency is a central challenge and a standard practice in recent video editing models to ensure temporal coherence.
>
> For example, FateZero [3] "to warp the middle frame for attribute and style editing" to guide the entire sequence, while Render-A-Video [4] applies "cross-frame attention to all sampling steps for global style consistency". Notably, both methods utilize a single text description as a global style command for all frames, which is conceptually similar to how our framework processes a single music feature stream as a global guide.
>
> [1] Deap: A database for emotion analysis; using physiological signals. IEEE transactions on affective computing, 2011.
>
> [2] LIRIS-ACCEDE: A video database for affective content analysis. IEEE Transactions on Affective Computing, 2015.
>
> [3] Fatezero: Fusing attentions for zero-shot text-based video editing. ICCV, 2023.
>
> [4] Rerender a video: Zero-shot text-guided video-to-video translation. SIGGRAPH Asia, 2023.
>
> **W2: Cross-attention**
>
> The core novelty of our multi-stream temporal ControlNet lies in demixing the audio into speech, effects, and music tracks. This allows user-provided audio to precisely control the generated video's lip motion, event timing, and visual mood, respectively. To achieve this, we design an interval stream for specific feature synchronization and a holistic stream for overall aesthetic presentation. As a result, our MTV framework achieves significantly better performance than the comparison methods (Tab. 2)
>
> Additionally, the injection module is more sophisticated:
>
> * The interval stream employs interval interaction blocks to model the interplay of speech and effects within each time interval using self-attention, which maintains coherence with inferred semantic features. These interacted features are then injected via the interval feature injection mechanism. Unlike standard cross-attention that applies a condition to all frames, our approach injects each audio feature only into its corresponding video time interval, enabling precise temporal synchronization.
>
> * The holistic stream uses a holistic context encoder to extract music features as video style embeddings. To ensure visual consistency, an average pooling is applied to merge all intervals. Subsequently, we estimate scale and shift factors from these holistic music features and inject them into all intervals via the holistic style injection formula (Eq. 3). Notably, no cross-attention is applied in this stream.
>
> **W3: Visual mood**
>
> We have adopted the Audio Consistency (Audio-C) metric, which uses ImageBind embeddings to calculate the cosine similarity between video and audio features. A higher alignment score indicates the generated video features are more semantically consistent with the audio. We further consider the FVD as an indirect indicator, which measures the distribution gap between generated videos and ground truth videos. A significant mismatched mood would result in a poorer FVD score.
>
> We further conduct an additional ablation study, where we remove the music track. As shown in the table below, this leads to a degradation in both FVD and Audio-C scores, demonstrating the important role of the demixed music track.
>
> If the reviewer finds any appropriate metric, we would be glad to add its quantitative results.
>
> | Method | FVD ↓ | Temp-C (%) ↑ | Text-C (%) ↑ | Audio-C (%) ↑ | Sync-C ↑ | Sync-D ↓ |
> | :--- | :---: | :---: | :---: | :---: | :---: | :---: |
> | W/o music track |695.88 |95.17 | 26.40 | 25.69 | 3.07 | **9.42** |
> | MTV | **626.06** | **95.40** | **26.55** | **26.22** | **3.17** | 9.43 |
>
> **Q1: Alternative audio encoders**
>
> In our empirical experiments, we find that using different audio encoders produces similar performance. This finding is consistent with previous Visual-Animation [6] (see its Appendix 6.7).
>
> We conduct an additional ablation experiment where we replace Wav2Vec with Beats for both the effects and music tracks. As shown in the table below, this ablation achieves comparable performance, suggesting that our current choice of Wav2Vec is a robust and effective one for this task.
>
> | Method | FVD ↓ | Temp-C (%) ↑ | Text-C (%) ↑ | Audio-C (%) ↑ | Sync-C ↑ | Sync-D ↓ |
> | :--- | :---: | :---: | :---: | :---: | :---: | :---: |
> | Beats | **598.53** | **95.91** | 26.25 | 25.28 | 3.02 | 9.52 |
> | Wav2Vec |  626.06 | 95.40 | **26.55** | **26.22** | **3.17** | **9.43** |
>
> [6] Audio-Synchronized Visual Animation. ECCV, 2024.
>
> **Q2: Mixed speech and sound effects**
>
> The speech and effect tracks are not mixed into a single input. Specifically, we apply a self-attention mechanism across the features of both the speech and effect tracks. This allows them to model their interplay at each time interval. However, they remain as two separate tracks after this step. Subsequently, these two tracks are injected into the video features independently using two separate masked cross-attention modules.
>
> We recognize that if two audio control signals are applied without any interaction, their independent features could create conflicting instructions and interfere with sensitive tasks like lip synchronization. Therefore, our goal is to construct a coherent control representation while retaining the distinct nature of each track. This is inspired by real-world scenarios where visual events are driven by a combination of sounds (e.g., a person speaking while their footsteps are heard). By modeling the interplay, our network can learn to produce a unified semantic feature that guides the larger video context (e.g., both lip movements and body posture).
>
> We further conduct an additional ablation study, where we remove the shared self-attention module that models the interplay and instead apply self-attention to each track individually. As shown in the table below, this ablation leads to a degradation in both lip-sync scores. Therefore, our interplay self-attention module is the solution to potential interference, instead of a source of it.
>
> | Method | FVD ↓ | Temp-C (%) ↑ | Text-C (%) ↑ | Audio-C (%) ↑ | Sync-C ↑ | Sync-D ↓ |
> | :--- | :---: | :---: | :---: | :---: | :---: | :---: |
> | W/o Share | 683.10 | 94.96 | 26.43 | 25.64 | 2.64 | 9.52 |
> | MTV | **626.06** | **95.40** | **26.55** | **26.22** | **3.17** | **9.43** |
>
> **Q3: Multiple speakers**
>
> In our specific implementation, we introduce an additional speech track for a second speaker during the multi-character training stage. The weights of the second speech track are initialized with those of the first speech track, which is pre-trained on a single-character subset. To handle multiple speakers, we perform speaker diarization using Scribe to distinguish active speaker segments during data processing (Line 124-126). After that, we crop the segments for the first and second speakers and feed the cropped audio into two individual speech tracks. This allows our MTV framework to handle speaking overlaps between multiple speakers.
>
> To map a speech track to its corresponding face region, we create a template to structure text descriptions so that the correspondence is adaptively learned (Line 203-210). These text descriptions explicitly state which speaker is active and their turn-taking order.
>
> As our model uses audio to control the video generation, the generated video is inherently aligned with the provided audio. Therefore, temporal misalignment does not occur.
>
> **Q4:  Three components**
>
> We demix audio into three components: speech, sound effects, and music. This methodology aligns with the standards of the CDX'23 challenge, and we adopt its leading models. In practice, this framework separates human voice into the speech track and instrumental accompaniment into the music track. All remaining audio components, including crucial modalities like ambient sounds and reverb, are thereby categorized as the effects track.
>
> To evaluate the information integrity of this process, we sum the three demixed audio tracks and calculate the L1 distance against the original audio. The table below presents the mean L1 distance, which demonstrates the high fidelity and information completeness of our separation.
>
> | Speech | Effects | Music | Speech+Effects+Music |
> | :--- | :---: | :---: | :---: |
> |0.008156 |0.019122 |0.017116 |**0.000863** |

---

> > ### Comment · Reviewer_uen2 · 2025-08-06
> > **Official Comment by Reviewer uen2**
> >
> > Thank you very much for the author's detailed reply. I still have some concerns.
> >
> > W1&W2: The author still needs to clarify whether using the average pooling layer to compress music features will damage the emotional expression of the video? What will be the result if the music features are also fused through cross-attention, and what is the performance?
> >
> > Another question about this issue is that compressing music features into a vector and learning this global emotional style is a bit like Emotion ID in Emotion TTS. However, will this approach hinder the generation of expressive audio and video?
> >
> > W3: If we look at Audio-C (%), removing the W/o music track does not drop much, which may not fully prove the part of effectiveness of "multi-stream temporal ControlNet". I admit that this approach (affine emotion with shift and scale) makes sense intuitively, but does it really work for emotional alignment during face expression change?
> >
> > In addition, a noteworthy issue is that the Sync-C and Sync-D in this paper seem to be relatively low? Because the Sync-C in most papers is 5.X~6.X[1][2], can the author explain the reason? Is this related to multiple speakers? Or is it the interference of background music? I want to know what the Sync-C and Sync-D of GT result are? Kindly remind, the author may need to check it.
> >
> > [1] SyncTalkFace: Talking Face Generation with Precise Lip-Syncing via Audio-Lip Memory
> >
> > [2] SyncTalk++: High-Fidelity and Efficient Synchronized Talking Heads Synthesis Using Gaussian Splatting

---

> ### Author Response · Authors · 2025-08-06
>
> Dear Reviewer, as the discussion period is ending soon, we would be grateful for any feedback on our rebuttal and are happy to answer any further questions.

---

> ### Author Response · Authors · 2025-08-07
> **Response to follow-up questions (1/2)**
>
> Thank you for thoughtful feedback on our work. We appreciate the opportunity to provide further clarification. We address each of your remaining points below and welcome any further discussion.
>
> **[W1 & W2]-1. Cross attention**
>
> We thanks for the reviewer's suggestion.  We have already begun running this ablation study, where music features are fused via cross-attention instead of our proposed holistic style injection. The experiment is currently in progress, and our preliminary observations support our original design choice.
>
> Interestingly, thanks to the powerful T2V backbone, we are not observing significant flickering in the generated video. The backbone itself is robust enough to maintain generation stability. Instead, the issue we have identified is that injecting music features at each timestep leads to a visual mood that feels disconnected from the overall feeling of the music. We believe this is because this per-interval approach lacks a global understanding of the audio's emotional tone. Instead, our original design using an average pooling layer effectively captures and provides this overall aesthetic presentation.
>
> We expect to post the final quantitative results from this ablation study here as a follow-up comment within the next 28 hours. We are sharing these preliminary findings now to ensure there is sufficient time to address any other questions you may have before the discussion period concludes.
>
>
> **[W1 & W2]-2. Expressive generation**
>
> To clarify, since the audio is a user-provided input, our model's goal is to generate a video that is expressively synchronized with it, not to generate the audio itself.
>
> Our framework is explicitly designed to "prevent music features from hindering the video generation" by decoupling the control streams (see Fig. 2 (b)). Specifically, the music track is processed in the holistic stream to establish the overall aesthetic presentation (i.e., the scene's mood), while the speech and effects are handled by the interval stream for interval-wise specific feature synchronization (i.e., lip motion and visual events). These two streams are not in opposition. For example, a character can be talking with an expressive smile (driven by the speech track) within a scene that has a globally gloomy mood (driven by the music track).
>
> Additionally, the foundational generation quality relies heavily on the pretrained T2V backbone (e.g., CogVideoX), as evidenced by the visual quality improvement when we integrated with Wan14B. The role of our MST-ControlNet is to ensure that the strong expressiveness of the backbone is accurately aligned with the audio.
>
> We will include these results in the final version.

---

> ### Author Response · Authors · 2025-08-07
> **Response to follow-up questions (2/2)**
>
> **[W3]-1. Audio-C**
>
> We agree that the drop in the Audio-C score for the "W/o music track" ablation is not significant. We believe this is expected and still demonstrates the effectiveness of our multi-stream design.
>
> The Audio-C metric primarily measures high-level semantic correspondence. When the music track is removed, the video's aesthetic mood may be less aligned, but the speech and effects tracks still ensure the primary events and lip movements are correctly synchronized . As the core semantic content remains correct, this maintains a reasonable Audio-C score.
>
> The resulting video is not "wrong", but rather adopts a more average presentation style. As a result, the score does not degrade significantly.
>
> **[W3]-2. Emotional alignment.**
>
> The affine transformation (holistic style injection) is intentionally not responsible for generating the dynamic facial expressions. This task is handled by our interval stream, which drives interval-wise facial muscle movements based on the speech track. Instead, the goal of the music track is to apply a consistent visual mood (e.g., color tone) globally across the frames.
>
> For instance, the speech track can drive a talking face with a smile, while the music track determines whether that smile appears in a warm context or gloomy one. This design allows our framework to align the global aesthetic presentation with the specific facial expression without conflict.
>
> **[W3]-3. Sync-C/D**
>
> We have verified that our Sync-C/D scores are calculated correctly. The perceived difference in scores stems from the significantly higher difficulty of our audio-sync cinematic video generation task compared to the talking head task in the cited papers.
>
> To provide a proper baseline, we present the scores on our test set below, including the Ground Truth (GT):
>
> | Method | MM-Diffusion | TempoTokens | Xing et al. | Ours (MTV) | Ground truth |
> | :--- | :---: | :---: | :---: | :---: | :---: |
> | Sync-C ↑ | 1.53 | 1.45 | 1.55 | 3.17 | **3.92** |
> | Sync-D ↓ | 11.21 | 10.48 | 10.50 | 9.43 | **8.34** |
>
> As shown, the GT score on our complex DEMIX dataset is already much lower than the 5.x-6.x scores reported on clean talking-head datasets. This is because:
>
> 1. Dataset complexity. Unlike standard talking-head datasets (e.g., VoxCeleb2) that use stable and front-facing heads, our DEMIX dataset contains "in-the-wild" cinematic videos. These clips often contain small or blurry face regions (e.g., Fig. 1, 1st row), and non-speaking clips where audio-visual speech correlation is naturally low (e.g., Fig. 1, 4th row, which has a Sync-C of only 1.26). These real-world challenges affect the SyncNet evaluation model itself (which is used to calculate Sync-C/D scores), lowering the scores even for ground truth videos.
>
> 2. Task difficulty. Methods like SyncTalk++ and SyncTalkFace are designed to animate a provided reference image or video, focusing solely on the head. In contrast, our framework generates the entire scene from only a text description, including the background, body, and face, without any visual reference. Therefore, a direct comparison of these scores can be misleading, as our framework solves a much broader and more challenging problem that existing talking-head methods do not support.

---

> ### Author Response · Authors · 2025-08-08
> **Experiment results for cross-attention ablation**
>
> Following up on our previous response, we have now completed the experiment regarding the reviewer's insightful suggestion of injecting music features using cross-attention. We present the quantitative results below:
>
> | Demix quality | FVD ↓ | Temp-C (%) ↑ | Text-C (%) ↑ | Audio-C (%) ↑ | Sync-C ↑ | Sync-D ↓ |
> | :--- | :---: | :---: | :---: | :---: | :---: | :---: |
> | CrossAttn | 656.22 | 95.38 | 26.48 | 25.76 | **3.19** | 9.47 |
> | MTV | **626.06** | **95.40** | **26.55** | **26.22** | 3.17 | **9.43** |
>
> These results lead to the following conclusions:
>
> 1. Video temporal consistency (Temp-C) shows only minor fluctuations, demonstrating that the T2V backbone effectively maintains generation stability regardless of the injection method.
>
> 2. Lip synchronization metrics (Sync-C/D) also show only minor fluctuations, providing the evidence that the music track is effectively decoupled from the speech and effects tracks, which independently control audio synchronization.
>
> 3. Overall video quality (FVD) shows a slight degradation, which suggests the visual mood generated by the cross-attention method deviates from the desired cinematic aesthetic.
>
> 4. The general audio-video alignment (Audio-C) also drops, indicating that the cross-attention approach provides less accurate audio-video correspondence overall.
>
> In conclusion, this ablation study confirms that while cross-attention is a valid mechanism, which may not be the optimal approach to inject music features.
>
> Additionally, we believe that our approach does not damage the emotional expression of the video. Compressing the music feature via average pooling provides a global understanding of the audio's emotional tone, which is ideal for establishing the overall aesthetic of a short video clip. As we detailed previously, this global visual mood does not interfere with the facial movements.  These interval-wise facial expression are driven independently by the speech track through our interval stream. This design ensures a clear decoupling between global visual mood and specific facial movements.

---

> > ### Comment · Reviewer_uen2 · 2025-08-09
> >
> > Thanks for the responses.
> > By the way, if the task is difficult, I think other aligning metrics should be considered to demonstrate synchronization performance.
> > Since GT Sync-C is around 3, it may not be the most ideal synchronization metric for fairness.
> >
> > Overall, most of my concerns have been addressed. I think this article is acceptable, but please make sure the related issues and discussions need to be clarified in the final version.
> >
> > Besides, I hope the author will make the source code and inference results of this work public to facilitate its development in the community.

---

> > > ### Author Response · Authors · 2025-08-09
> > >
> > > We sincerely thank you for detailed and constructive feedback throughout this entire process, and for your final positive assessment. Your rigorous questions have been invaluable in helping us significantly strengthen the paper.
> > >
> > > We agree with your valuable suggestion regarding the evaluation metrics. We will incorporate all these important discussions and our new results into the final version. We also plan to release our source code and models to the community upon acceptance.
> > >
> > > Thank you again for your time and guidance.

---

### Official Review · Reviewer_453Q · 2025-07-03

**Clarity:** 2
**Significance:** 2
**Originality:** 2
**Rating:** 4
**Confidence:** 3

**Summary:**

This paper introduces MTV framework (Multi-stream Temporal control for audio-sync Video generation), a framework for generating high-fidelity cinematic videos with precise audio-visual synchronization across diverse audio types. Existing methods struggle with under-specified audio-visual mapping and inaccurate temporal alignment. MTV addresses these challenges by separating audio inputs into speech, effects and music tracks, for disentangled control over lip motion, event timing, and visual mood, enabling fine-grained and semantically aligned video generation. To support this framework, the authors present an audio-sync video generation dataset, DEMIX, structured into five overlapped subsets and introduce multi-stage training strategy. Additionally, the paper proposes MST-ControlNet (Multi-Stream Temporal ControlNet) to distinctively process demixed audio stream and inject audio features of each track into corresponding time intervals, achieving fine-grained synchronization. The performance of MTV for video quality, text-video consistency, and audio-video alignment outperforms 3 baseline methods across six standard metrics.

**Questions:**

- Robustness to imperfect demixing isn't evaluated. Have you evaluated synchronization quality under varying levels of demixing noise?
- Can MTV maintain its performance gain when it is applied on various audio-visual datasets or other state-of-the-art video-diffusion models? (See weakness 2)

**Ethical Concerns:**

["NO or VERY MINOR ethics concerns only"]

**Final Justification:**

After thoroughly reviewing the authors' rebuttal, I find that they have sufficiently addressed most of the key concerns raised in the initial review. They demonstrated the generalizability of the proposed MTV framework through additional results on new datasets (e.g., Landscape, Audioset-Drum) and a different backbone (Wan14B). They also validated robustness to varying demixing quality and clarified concerns on resolution and video length by providing detailed qualitative examples. In addition, they presented ablation studies and justifications for major design choices, making the method more convincing. I would kindly encourage the authors to reflect these additional experiments and clarifications in the camera-ready version to further strengthen the quality of the paper. Plus, I am happy to increase my rating to a more positive one.

**Limitations:**

- MTV generates audio-sync videos only at a 480 x 720 resolution and 49-frame segments, limiting its scalability to high-resolution or long-form video generation.
- The paper lacks ablations or empirical justification for key design choices (e.g., stream interaction blocks, training schedule).

**Quality:**

3

**Strengths And Weaknesses:**

Strengths
- MTV method is directly aligned with the two core motivations--under-specified audio-visual mapping and inaccurate temporal alignment--by explicitly demixing audio into distinct controlling tracks with dedicated temporal control mechanisms.
- The authors contribute DEMIX, a large-scale cinematic video clips with demixed audio tracks, structured into five overlapping subsets, which facilitates multi-stage training.

Weaknesses
- MTV's performance depends on the accuracy of upstream demixerrs (MVSEP, Spleeter). Errors in source separation can propagate and degrade synchronization.
- Quantitative results are reported only on DEMIX dataset and CogVideoX baseline model. It remains unclear how well MTV generalizes to other T2V models and different datasets (e.g., non-cinematic content)
- The paper relies solely on automated metrics; incorporating qualitative human assessments would better validate diverse aspects including detailed semantic consistency, narrative fluency of video, and user preferences.

---

> ### Author Rebuttal · Authors · 2025-07-30
>
> We thank the reviewers for their insightful and constructive feedback.
>
> In accordance with the rebuttal policy, we are unable to include new qualitative results (e.g., generated videos) or external links. We assure the reviewers that, should the paper be accepted, the camera-ready version will be thoroughly updated to include:
>
> 1. All additional experiments and qualitative results are discussed in this rebuttal.
>
> 2. A public link to an open-source demo showcasing our MTV framework as a practical video generation tool.
>
> We now address the specific points raised by the reviewers.
>
> ## ***Weakness***
>
> **W1: Errors in source separation**
>
> We agree that the upstream source separation is not always perfect. However, all of our reported results are generated from these imperfectly separated audio tracks, not from idealized clean inputs. As shown in Tab. 2 and Fig. 3, our framework achieves state-of-the-art performance across six standard metrics, directly demonstrating its inherent robustness in real-world scenarios. This stems from our demixing filtering strategy (Lines 117-122). We explicitly filter out extremely poor-quality samples during training, which enables our model to learn precise audio-visual synchronization,  while the remaining unclear separations serve as a form of data augmentation that enhances robustness.
>
> To completely bypass this separation challenge, we further developed a demo. It uses Qwen3 to interpret user prompts into audio descriptions, synthesizes them into perfectly clean audio with ElevenLabs, and then generates a synchronized video with our MTV framework. This presents a complete text-to-video pipeline to avoid potential source separation errors entirely.
>
> **W2: Additional datasets and backbone**
>
> We thank the reviewer for this valuable suggestion.
>
> 1. Evaluation on new datasets
>
> To demonstrate the generalization capabilities of our MTV framework, we follow TempoTokens to conduct additional experiments on both the Landscape and AudioSet-Drum datasets. To make a fair comparison, we fine-tune all baseline methods (MM-Diffusion, TempoTokens, and Xing et al.) on our DEMIX dataset using their official training schedules, and evaluate them on these separate datasets. As shown in the tables below, our MTV framework still achieves significantly better performance.  Since neither dataset includes human talking, the lip synchronization metrics (Sync-C and Sync-D) are not applicable for this evaluation.
>
> #### **Comparison on Landscape datasets**
>
> | Method | FVD ↓ | Temp-C (%) ↑ | Text-C (%) ↑ | Audio-C (%) ↑ |
> | :--- | :---: | :---: | :---: | :---: |
> | MM-Diffusion | 807.65 | 94.74 |14.66| 16.59 |
> | TempoTokens | 797.33 | 94.67 | 21.73 | 18.86 |
> | Xing et al. | 838.03 | 94.71 | 21.04 | 18.70 |
> | Ours (MTV) | **697.51** | **96.98** | **25.35** | **23.37** |
>
> #### **Comparison on AudioSet-Drum dataset**
>
> | Method | FVD ↓ | Temp-C (%) ↑ | Text-C (%) ↑ | Audio-C (%) ↑ |
> | :--- | :---: | :---: | :---: | :---: |
> | MM-Diffusion | 1520.09 | 94.59 | 14.90 | 14.11 |
> | TempoTokens  | 1512.97 | 94.28 | 23.18 | 15.59 |
> | Xing et al. | 1589.46 | 94.49 | 23.73 | 17.84 |
> | Ours (MTV) | **1511.53** | **97.50** | **25.62** | **39.61** |
>
> 2. Adaptability to additional T2V backbone
>
> Our proposed MSTControlNet can be seamlessly integrated into various T2V backbones (e.g., OpenSora and Wan14B) without architectural changes, as they share a similar DiT-based structure. Due to time limitations, we only implement and test the integration with Wan14B. Specifically, our interval feature injection and holistic style injection modules are added after each text cross-attention layer.
>
> For inference, MTV (Wan14B backbone) requires 180s to generate an 81-frame, audio-synchronized video at a 480P resolution on 8 NVIDIA A100 GPUs. We report the new quantitative results below.
>
> | Method | FVD ↓ | Temp-C (%) ↑ | Text-C (%) ↑ | Audio-C (%) ↑ | Sync-C ↑ | Sync-D ↓ |
> | :--- | :---: | :---: | :---: | :---: |:---: |:---: |
> | CogVideoX |626.06 |95.40 |26.55 |26.22 | **3.17** | **9.43** |
> | Wan14B | **353.61** | **96.36** | **27.23** | **26.49** | 3.08 | 9.56 |
>
> **W3: Human assessments**
>
> To better evaluate our method from a human perception perspective, we conducted three subjective user study experiments as requested. We presented videos generated by our method and all baselines to participants and asked them to choose the best one based on the following criteria:
>
> * Semantic consistency: "Which video best matches the text description?"
>
> * Motion fluency: "Which video displays more realistic and temporally coherent motion?"
>
> * Overall preference: "Which video do you prefer overall, considering all aspects?"
>
> For each study, we randomly select 50 text descriptions from the test set, and the evaluations are conducted by 25 volunteers. The table below shows the percentage of times each method is chosen as the winner. Our method is consistently favored by human observers and has achieved the highest scores across all three subjective criteria.
>
> We hope that our new experimental results have fully addressed the reviewer's concerns, and we would be happy to provide further clarification on any remaining points during the author-reviewer discussion period.
>
> | method \ metrics | Semantic consistency ↑ | Motion fluency ↑ | Overall preference ↑ |
> | ---------------- | -------------------- | -------------- | ------------------ |
> | MM-Diffusion     | 0.96%   | 0.64%   | 0.72%  |
> | TempoTokens      | 13.60%     | 8.96%   | 12.00% |
> | seeing and hearing| 11.28%     | 12.56%   | 12.40% |
> | Ours(MTV)        | **74.16%**     | **77.84%** | **74.88%**  |
>
>
> **Q1: Varying levels of demixing quality**
>
> We follow the reviewer's suggestion to conduct an analysis of our model's performance under varying levels of demixing quality. We create three quality-based subsets (Level 1 being the highest quality, Level 3 the lowest), with each subset containing 20 samples. The quality level for each sample is determined by a consensus rating from 5 human volunteers. We also evaluate a "Random" subset, which is randomly sampled from the entire test set for baseline comparison.
>
> As the results below show, our model is robust to the quality of the demixed audio. Its performance on metrics like FVD and Text-C shows only minor fluctuations across the quality levels, while audio-visual correlation metrics (Audio-C and Sync-C) are even slightly better on lower-quality inputs. This may suggest that our model effectively focuses on audio cues that persist even when the demixed audio is not perfectly clean. Notably,  with any-level quality inputs, our model remains superior to all baselines evaluated on the Random subset. Therefore, our multi-stream approach provides robust gains overall.
>
> | Demix quality | FVD ↓ | Temp-C (%) ↑ | Text-C (%) ↑ | Audio-C (%) ↑ | Sync-C ↑ | Sync-D ↓ |
> | :--- | :---: | :---: | :---: | :---: | :---: | :---: |
> | MTV (Level1) | **617.23** | 95.14 | **27.03** | 24.81 | 2.88  | **8.99** |
> | MTV (Level2) | 665.18 |95.60 |26.66 | 26.01 | 3.13 | 9.29 |
> | MTV (Level3) | 638.92 | **95.88** | 26.52 | **26.87**| **3.46** | 9.41 |
> | MTV (Random) | 626.06 | 95.40 | 26.55 | 26.22 | 3.17 | 9.43 |
> | MM-Diffusion (Random) | 879.77 | 94.15 | 15.61 | 5.43 | 1.53 | 11.21 |
> | TempoTokens (Random) | 795.88 | 93.13 | 24.68 | 6.71 | 1.45 | 10.48 |
> | Xing et al. (Random) | 805.23 | 93.30 | 24.51 | 7.30 | 1.55 | 10.50 |
>
> **Q2:  Additional datasets and backbone**
>
> See Reviewer 453Q-W1.
>
> **L1: Resolution and duration**
>
> In fact, the resolution of our MTV framework ($480 \times 720$) is the highest among all comparison methods. For instance, MM-Diffusion is limited to a $256 \times 256$ resolution, while TempoTokens and Xing et al. can only generate $384 \times 384$ videos. Additionally, as demonstrated in our previous response (W2), we have adapted our MSTControlNet to the Wan14B backbone, enabling our framework to generate high-resolution $720 \times 1280$ videos.
>
> Our framework is fully capable of generating long videos. Numerous examples of long videos are already present in our submission. For instance:
>
> * In the main paper, the video in the 2nd row of Fig. 1 is 132 frames, and the video in the 2nd row of Fig. 5 is 146 frames.
>
> * The supplementary material contains even longer examples, including videos of 110, 116, 156, and up to 234 frames.
>
> **L2: Key design choices**
>
> The architecture of our stream interaction block is inspired by the dual-stream DiT block of the HunyuanVideo, a proven effective architecture. Beyond this, we already provide extensive ablation studies in the paper:
>
> * We analyze the architecture of our MTSControlNet in Sec. 5.2 (Lines 250-255).
>
> * We study the optimal number of blocks in Appendix A.2.
>
> We further conduct an additional ablation study ("Linear"), where we replace our interval interaction block with a stack of linear layers (following Hallo3). As the results shown below, our proposed interval interaction block achieves superior performance on this task.
>
> We also conduct an additional ablation study ("Mix") to validate our training strategy, where we train our MTV framework on the full DEMIX dataset, without the multi-stage training strategy.
>
> As the results below show, this ablation significantly degrades lip synchronization performance. We believe this is because lip-sync is a highly sensitive task that requires precisely aligning audio with a small visual region (i.e., the mouth). Without first learning this fundamental alignment on a simpler subset of facial data, the model struggles to learn this correspondence when faced with the complexity of large motions and multiple people.
>
> | Demix quality | FVD ↓ | Temp-C (%) ↑ | Text-C (%) ↑ | Audio-C (%) ↑ | Sync-C ↑ | Sync-D ↓ |
> | :--- | :---: | :---: | :---: | :---: | :---: | :---: |
> | MTV (Linear)  | 644.72 | 95.15 | 26.51 | 26.14 | 2.19 | 9.56 |
> | MTV (Mix) | 653.14 | 94.63 | 26.27 | 24.30 | 2.16 | 9.59 |
> | MTV | **626.06** | **95.40** | **26.55** | **26.22** | **3.17** | **9.43** |

---

> ### Author Response · Authors · 2025-08-06
>
> Dear Reviewer, as the discussion period is ending soon, we would be grateful for any feedback on our rebuttal and are happy to answer any further questions.

---

> ### Author Response · Authors · 2025-08-07
>
> Dear Reviewer 453Q,
>
> With the discussion period ending soon, we would be grateful for any final thoughts on our rebuttal. We are happy to clarify any remaining points.
>
> Thank you for your time and consideration.

---

> > ### Comment · Reviewer_453Q · 2025-08-08
> >
> > After thoroughly reviewing the authors' rebuttal, I find that they have sufficiently addressed most of the key concerns raised in the initial review. They demonstrated the generalizability of the proposed MTV framework through additional results on new datasets (e.g., Landscape, Audioset-Drum) and a different backbone (Wan14B). They also validated robustness to varying demixing quality and clarified concerns on resolution and video length by providing detailed qualitative examples. In addition, they presented ablation studies and justifications for major design choices, making the method more convincing. I would kindly encourage the authors to reflect these additional experiments and clarifications in the camera-ready version to further strengthen the quality of the paper. Plus, I am happy to increase my rating to a more positive one.

---

> ### Author Response · Authors · 2025-08-08
>
> We sincerely thank the reviewer for their positive feedback and for raising their score.
>
> We will incorporate all of these additional experiments and clarifications into the final version.

---

### Note · Authors · 2025-08-12

We sincerely thank the Area Chair and all four reviewers for a rigorous and highly constructive review process. Following the instruction, we now provide a summary of the discussions and their conclusions below.

* Reviewer 453Q initially raises concerns regarding the robustness to varying demixing quality, results on public datasets, performance with different pre-trained components, and justification for key design choices. After we conduct comprehensive experiments addressing these points, the reviewer concludes that we "have sufficiently addressed most of the key concerns" and are "happy to increase my rating to a more positive one"

* Reviewer uen2 initially raises concerns regarding our music injection module, evaluation mertics for visual mood, and alternative technical designs. After we explore approaches they suggested and provide a deatiled analysis for the evaluation metrics, the reviewer concludes that "most of my concerns have been addressed" and "this article is acceptable"

* Reviewer 3d1R initially raises concerns similar to those of Reviewer 453Q, while additionally questioning our comparison fairness, the universality of the training schedule, and the model's ability to learn from imperfect audio.  After we provide additional experiments result and detailed explainations, the reviewer confirms our responses have "fully resolved my remaining questions" and has "raised my score from 2 to 4"

* Reviewer 1iMS, while finding "no crucial weaknesses regarding the design of the proposed method and the experiments", raises a concern regarding data copyright and suggests an alternative layer-wise injection approach. After we provide a detailed clarification on the copyright and conduct the requested ablation study, the reviewer  states that "I have no further questions" and they would "like to maintain my rating", which is positive from the start

* All four reviewers identify our proposed DEMIX dataset is a key strength, which "is a solid contribution to the community" (3d1R) and "would benefit a lot the community" (1iMS)

We are confident that these additions have significantly improved the paper's quality and contribution.  We will incorporate all of these valuable discussions and new results into the final version. We will also release the code, model, and dataset to further benefit the community upon acceptance.

Thank you for your time and consideration.

---

### Decision · Program_Chairs · 2025-09-17

**Decision:**

Accept (poster)

**Comment:**

Strengths
* DEMIX dataset pairing video and separated audio tracks (speech, effects, and music)
* Demixing and curation strategy for the dataset
* (debatable - see Weaknesses) Multi-Stream Temporal ControlNet with time interval injection for speech and effects, holistic style injection for music
* Results are convincing

Weaknesses
* Condition injection with a ControlNet-like architecture is common

During the rebuttal, the authors conducted many additional experiments, such as using Wan instead of CogVideoX as a T2V backbone and varying demixing quality, based on the reviewer's comments. These additional experiments will strengthen the paper with additional insights for the community.

Therefore, I recommend accepting the paper.